# Malnutrition among under-five children in amhara and oromia regions, Ethiopia: Continuous time markov multi-state modeling

Dafa Duge Wachifo[1]*, Dereje Danbe Debeko[1], Zeytu Gashaw Asfaw[2,3]

1 Department of Statistics, Hawassa University, Ethiopia, 2 Department of Epidemiology and Biostatistics, Addis Ababa University, Ethiopia, 3 Institute of Medical Biometry and Statistics, Faculty of Medicine and Medical Centre, University of Freiburg, Freiburg, Germany

* dafaduge@gmail.com

## Abstract

### Background

Ethiopia faces a high burden of undernutrition prevalence, ranking among the 15 worst-affected nations globally. So, this study aimed to find out how often and how long under-five children (U5C) in Ethiopia's Amhara and Oromia regions move between different states of Composite Index Anthropometry Failure (CIAF) as well as what factors influence these changes.

### Methods

The data used for this study was extracted from the International Food Policy Research Institute. The institute conducted a follow-up survey in three consecutive rounds: February 8, 2018–April 25, 2018; July 25, 2019–October 23, 2019; and February 8, 2021–April 25, 2021, respectively. The inclusion criteria were households that had children between the ages of 0 and 35 months, were participants in the safety net program, and had the mother or primary female caregiver during the baseline survey. A total of 3,044 households having children with at least two complete anthropometric measurements were included. A continuous-time multi-state Markov model was used to estimate transitions and their probability between CIAF's states.

### Results

Nourished children had a 71% probability of becoming undernourished. Time taken to recover from undernourished state for U5C was 41 months on average. 75% of the U5C's life is spent in the undernourished state. Girls had a 1.824 times higher likelihood of recovering from an overnourished state and were less likely to transit from a nourished state to an undernourished state compared to boys (HR: 0.8013). Children older than two years were more likely to recover from undernourished and overnourished states respectively (HR: 1.013 & 1.036), to a nourished state and less likely to

**Data availability statement:** https://dataverse.harvard.edu/dataset.xhtml?persistentId=doi:10.7910/DVN/MBRDZ7.

**Funding:** The author(s) received no specific funding for this work.

**Competing interests:** The authors declare that they have no known competing financial interests or personal relationships that could have appeared to influence the work reported in this paper.

**Abbreviations:** AIC: Akaike Information Criterion; BMI: Body Mass Index; CARE: Cooperate Assistance and Relief Everywhere; CIAF: Composite Index Anthropometry Failure; CI: Confidence Interval; Df: Degrees of Freedom; EPHI: Ethiopia Public Health Institute; GO: Governmental Organization; HAZ: Height for Age Standard |Score; HMM: Homogeneous Markov Models; HR: Hazard Ratio; IFPRI: International Food Policy Research Institute; IPV: Intimate Partner Violence; IRB: Institutional Review Board; ML: Maximum Likelihood; MLE: Maximum Likelihood Estimator; MSM: Multi State Models; CTMSM: Continuous-Time Markov Multistate Model; NHM: Non-Homogenous Model; NGO: None Governmental Organization; ORDA: Organization for Rehabilitation and Development in Amhara; PSNP: Productive Safety Net Program; RERC: Research Ethics Review Committee; SD: Standard Deviation; SDG: Sustainable development goal; SPSS: Statistical Package for Social Science; SSA: Sub-Saharan Africa; U5C: Under five children; UN: United Nations; UNICEF: United Nations Children's Fund; WAZ: Weight for age standard score; WHO: World Health Organization; WHZ: Weight for height standard score.

transit back to the malnutrition states (HR: 0.9693 & 0.9662). Children of educated mothers and residents in the Oromia region had lower risk of transition from a healthy to an undernourished state respectively (HR: 0.8171& 0.8074).

## Conclusion

Undernutrition will affect most U5Cs. Children whose mothers had no education and live in the Amhara region are more susceptible to undernutrition. The Ministry of Health and other relevant stakeholders should develop a practical intervention to enhance adult and maternal education programs.

---

## 1. Introduction

Undernutrition can manifest in four forms: wasting, stunting, underweight, and micronutrient deficiencies. These conditions can lead to irreversible physical and cognitive damage, recent rapid weight loss-related risk of death, and reduced growth-development. However, overweight is the result of an excess intake of energy or nutrients, which can cause increased risk of diet-related non-communicable diseases in later children's lives [1,2]. Undernutrition is linked to 45% of deaths among under-five children (U5C), primarily in sub-Saharan Africa [3]. In 2022, the global prevalence of stunting among U5C was 22.3%, with 43% of the affected children living in Africa. Ethiopia had a particularly high prevalence of stunting, at 34.4%, which is classified as "very high" according to the WHO-UNICEF Technical Advisory Group on Nutrition Monitoring thresholds established in 2018. This rate is higher than that of East Africa's average (30.6%) and most neighboring countries such as Somalia (18%), Kenya (18.4%), Djibouti (18.7%), and South Sudan (27.9%)—except for Eritrea (50.2%) and Sudan (36%) [1,4]. Similarly, the global prevalence of wasting among U5C was 6.8%. In Ethiopia, the prevalence of wasting was equal to the global average at 6.8%, while the underweight rate stood at 21%. With these figures, Ethiopia is better than those of most neighboring countries, except for Kenya (5% wasting and 10% underweight). In terms of overweight, the global prevalence among U5C was 5.6%. Ethiopia reported a lower prevalence of 2.7%, which is below both the global average and the rates in all neighboring countries [4,5].

Ethiopia's undernutrition rate has slightly decreased from 38% of stunting, 10% of wasting, and 24% of underweight in 2016 to 34.4% of stunting, 6.8% of wasting, and 21% of underweight in 2022. But it still lags behind the Sustainable Development Goal 2 (SDG-2) of ending all forms of undernutrition by 2030 [5–7]. The UN agencies were calling for urgent action on child wasting in their meeting, Ethiopia listed among the 15 worst-affected countries in the world. From these countries, over 30 million children were suffering from acute malnutrition; of these 8 million were severely wasted. Most of the listed 15 countries, such as Afghanistan, Burkina Faso, Chad, DRC, Ethiopia, Haiti, Kenya, Madagascar, Mali, Niger, Nigeria, Somalia, South Sudan, Sudan, and Yemen were from Africa except Haiti, Afghanistan, and Yemen. Moreover, Kenya, South Sudan, Sudan, and Somalia are neighboring countries of Ethiopia [8].

Previous studies in Ethiopia have attempted to identify determinants of malnutrition using conventional indicators, such as, stunting, wasting, or underweight separately [9–14]. However, these studies neither aggregated the indexes nor considered their transient or dynamic nature. These indicators partly overlap and therefore fail to present an accurate and convincing estimate of the total proportion of malnourished children in the population [15,16]. The conventional indicators of malnutrition such as stunting, wasting, and underweight separately fail to offer a comprehensive assessment of the overall burden of malnutrition among children. This limitation arises because children can experience multiple forms of malnutrition at the same time.

Using the composite index anthropometry failure (CIAF) approach is more effective than using conventional indicators, as it captures multiple forms of malnutrition simultaneously and provides a convincing estimate of the overall burden in the population [16]. Previous studies conducted in Ethiopia using the CIAF have not accounted for the dynamic nature of malnutrition, nor have they incorporated overnutrition into the composite index [17,18]. A recent study sought to capture the dynamic characteristics of stunting by utilizing longitudinal data and applying a multi-state model. However, the study did not incorporate other indicators of malnutrition [19]. Other studies that used longitudinal data from the International Food Policy Research Institute (IFPRI) tried to examine the influence of poultry transfers on diet diversity, the effectiveness of an aspiration intervention of telling stories of improved households experiences with other rural households for successful escapes from poverty, and whether and how the Ethiopian government's public works program or strengthen productive safety net program institute resilience (SPIR)—which involves food or cash transfers for seasonal labor—along with complementing activities that involve both men and women—affected intense partner violence (IPV) [20–22]. However, these studies have not explored the implications of the intervention on composite measures such as the CIAF. Therefore, further research is needed to evaluate the intervention's impact on children's nutritional status over time. To the best of our knowledge, existing studies have not comprehensively incorporated the CIAF, as they have typically excluded either the dual burden of undernutrition and overnutrition, or the dynamic aspects of malnutrition among U5C, particularly with a continuous-time multistate Markov model. Therefore, this study aimed to find out how often and how long U5C move between different states of CIAF as well as what factors influence these changes in Ethiopia's Amhara and Oromia regions. The CIAF reflects a transient or dynamic nature, which necessitates analysis using a multi-state model (MSM). As a composite measure, the CIAF aggregates multiple forms of malnutrition, including stunting, wasting, underweight, and overweight, and serves as a tool to track overall nutritional status. A children's status may transit between different CIAF states or recover to a healthy state over time, highlighting its dynamic characteristics. Therefore, the MSM is well-suited for analyzing such complex data structures, as it enables a comprehensive understanding of how children transition between various states, such as undernourished, nourished, or overnourished, or remission. Therefore, this study could help to simplify interpretations of conventional indicators by providing a single conclusion on the overall burden of malnutrition. It could also provide estimates for transition rates, transition probabilities, and driving factors at specific times between different states of CIAF. In addition, it could contribute to the application of MSM in nutrition.

## 2. Method

### 2.1 Study population

The target population of this study was U5C from households participating in the PSNP in the Amhara and Oromia regions of Ethiopia. As the survey did not include other regions of the country, the findings of this study are limited to these two regions.

### 2.2. Research design and data source

The dataset was obtained from the IFPRI. To improve outcomes related to livelihoods, food security, child nutrition, women's empowerment, mental health, and intimate partner violence (IPV), IFPRI conducted an experimental, quantitative impact evaluation of the *Strengthen PSNP Institute and Resilience* (SPIR) program. This evaluation was designed

to measure the causal impact of a multi-sectoral "graduation model" that integrates interventions in livelihoods, nutrition, gender equity, and mental health [23]. To support the implementation of this program, World Vision and its partners, Cooperative for Assistance and Relief Everywhere (CARE) and Organization for Rehabilitation and Development in Amhara (ORDA), have been providing services to more than 500,000 PNSP clients in 15 food-insecure districts in the Amhara and Oromia regions. This effort is funded by USAID's Bureau for Humanitarian Assistance and carried out in close collaboration with the Government of Ethiopia [21–23]. Initially, 196 kebeles from Amhara (115 kebeles) and Oromia (81 kebeles) were selected from 15 PSNP implemented districts. However, four kebeles (3 from Amhara and 1 from Oromia) were dropped due to no PSNP clients in two kebeles, and there was ongoing civil unrest in two other kebeles. Thus, the evaluation sample comprises 192 kebeles. From each kebele, 18 households were randomly selected, and then a total of 3,314 households that satisfied the inclusion criteria were included. The inclusion criteria were that households be a PSNP client, have at least one child aged 0–35 months at baseline, and have the indexed child's (0–35-month-old) biological mother or primary female caregiver as a member of the household among the eligible households. Then the baseline, midline, and endline surveys were conducted from February 8 to April 25, 2018; July 25 to October 23, 2019; and February 8, 2021, to April 25, 2021, respectively. The standardized questionnaires were used for data collection. The mother/ primary female caregiver questionnaire constitutes the nutritional practices for the indexed children and their anthropometric measures. Children whose ages were less than 36 months were designated as the baseline indexed children [23]. We accessed the required data from IFPRI according to the data access request and adhered to their data-sharing protocols. Finally, the data set was accessed on May 29, 2023. Based on the current study's criteria, a total of 3,044 households having at least two complete anthropometric measurements for indexed children at the baseline were included in the study.

### 2.3. Ethical statements

IFPRI received approval from its Institutional Review Board (IRB) at baseline, and it was updated for the second-round survey. IFPRI also received ethics approval from the Institutional Review Board (IRB) at Hawassa University. Additionally, IFPRI had collected informed oral consent from all participants prior to the start of any interview. The entire field team was trained on ethical data collection, prior to the start of any interviews. Before beginning a survey, enumerators read each respondent a brief description of the study that was being conducted, informed them that their participation in the study was voluntary and that they could discontinue participating at any time, and asked whether they agreed to respond to the interview questions. The enumerator only completed a survey if they received verbal consent from the target respondent to participate in the study. Anonymized versions of the datasets that exclude personal identifiers were made available for public access. We have also received ethical clearance from the Hawassa University College of Natural and Computational Sciences Research Ethics Review Committee (RERC). Thus, no risk of harm is posed to the study participants.

### 2.4. Study variables

**2.4.1. Dependent variable.** The dependent variable in this study was CIAF among U5C which is categorical. To minimize measurement errors, anthropometric data were collected using standardized procedures. The height/length, mid-upper arm circumference (MUAC), and weight of the indexed child and mother/primary female caregiver were measured twice. If the differences between the two measurements were less than 0.01 cm for height/length and MUAC and 0.01 kg for weight, the measurements were considered accurate. If the differences exceeded these thresholds, a third measurement was taken and recorded. Standard scores were computed using the zscorer R package for three anthropometric indicators: height-for-age z-score (HAZ), weight-for-height z-score (WHZ), and weight-for-age z-score (WAZ). The cut-off points for undernutrition were based on the WHO Child Growth Standards median: stunting is defined as HAZ < −2 standard deviations (SD), wasting as WHZ < −2 SD, underweight as WAZ < −2 SD, and overweight as WHZ > +1 SD [1,2]. To assess the overall nutrition status of U5C, the study employed the revised and revisited CIAF as

the primary outcome variable [16]. They stated, as Svedberg in 2000 has pointed out, that stunting, underweight, and wasting are dependent entities where WAZ is often used to reflect the extent of both chronic and acute malnutrition. However, WAZ cannot distinguish between children who are with HAZ and/or WHZ due to overlapping. So, it provides an underestimate of the extent of anthropometric failure in a population; hence, CIAF is recommended as a solution [16]. The CIAF status was computed by using nine combinations of four indexes, as shown in Table 1 and Fig 1. Then it was categorized into three categories: undernourished (B-F or Y), nourished (A), and overnourished (G or H). Simplifying the CIAF into three distinct categories can reduce ambiguity from overlapping subcategories and make the results more interpretable.

*The revised formula for* det *ecting*
$Under\_nourished(U) = B + C + D + E + F + Y, \quad Nourished(N) = A, \quad Over\_nourished(O) = G + H$
$Hence, CIAF\ categorized\ as\ :\ U = State\ 1,\ N = State\ 2,\ and\ O = Satate\ 3$

**2.4.2. Independent variables.** The inclusion of explanatory variables was guided by three main criteria: the aim of the study, the availability of variables in the dataset, and recommendations from existing literature as key predictors of

**Table 1. Proposed Composite Index of Anthropometric Failure Categories.**

| CIAF Categories | Wasted | Stunted | Underweight | Overweight |
|---|---|---|---|---|
| *Group A – No failure* | No | No | No | No |
| *Group B – Wasted only* | Yes | No | No | No |
| Group C – Wasted & Underweight | Yes | No | Yes | No |
| Group D – Wasted, Stunted & Underweight | Yes | Yes | Yes | No |
| Group E – Stunted & Underweight | No | Yes | Yes | No |
| Group F – Stunted only | No | Yes | No | No |
| Group G – Stunted & Overweight | No | Yes | No | Yes |
| Group H – Overweight only | No | No | No | Yes |
| Group Y – Underweight only | No | No | Yes | No |

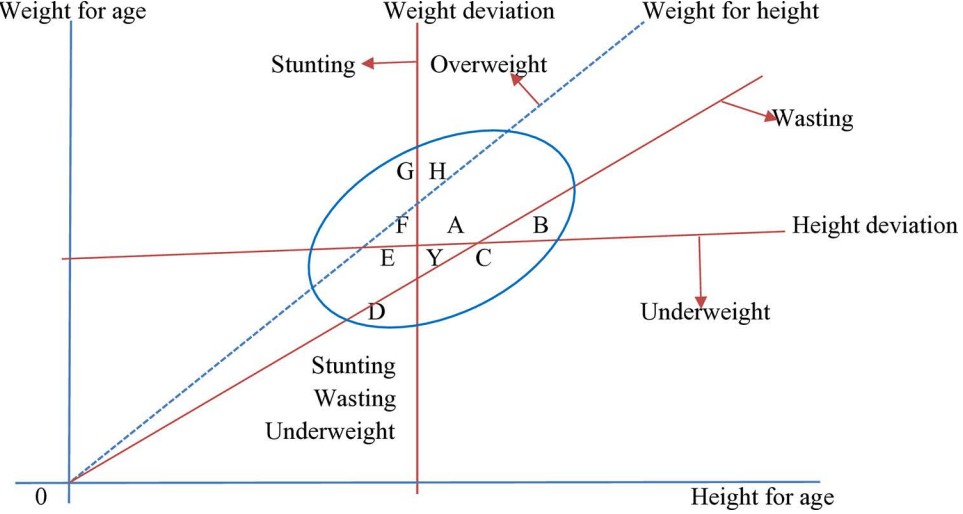

**Fig 1. Revised and Revisited CIAF with nine sections [16].**

child malnutrition. The inclusion of covariates was not exhaustive but rather focused on variables most relevant to the study's aim and feasible within the available data. Based on these considerations, the following covariates were included. **Child characteristics**: sex, age, birth weight, illness episodes, breastfeeding duration, and immunization status. **Mother characteristics:** age, marital status, highest education level, employment status, feeding practices, childcare activities, exposure to health, and nutrition services. **Household characteristics:** family size, sex of the household head, religion, zone, and region.

### 2.2 Method of data analysis

#### 2.2.1 Definition and assumption of multi-state model.

A multistate model is a framework that uses continuous-time processes to describe and model subjects' experiences over a time course. All multistate models consist of two essential components: the states and the transitions. "State" is the time-varying/longitudinal status of a subject at a given time. "Transition" is a directional movement from one state to another. A state can be transient or terminal. A state is considered transient if a transition from that state to another state is possible, whereas a state is considered terminal (absorbing) if a transition from that state to another state is not possible; that is, once a subject enters a terminal state, s/he is assumed to remain permanently in that state [24]. In biomedical applications, the states may be health conditions, disease stages, or a nonfatal complication in the course of an illness [25]. In time-to-event data, Kaplan–Meier estimators and Cox proportional hazard models are adequate to use in studies where there is only one type of event of primary interest. Competing risks models can accommodate transition between one initial state and several mutually exclusive absorbing states. The recurrent event models are appropriate when transitioning from one state to another state, either recurrent event 1 or event 2, or so on. However, when there are multiple events of interest, these methods may not provide a full picture of the relationship. In such a case, the multistate model provides a flexible and broader framework to extend familiar methods. When the process involves transitions between several well-defined distinct states, in such cases the multistate model is appropriate over other statistical models [24]. The progression of the nutritional status of U5C is an example of complex processes with intermediate events. Therefore, the multistate model is preferred over other statistical models in this specific case.

In this specific case, a multistate model described how a child's nutritional status or CIAF moves between a series of its states. CIAF has three states: undernourished (U), nourished (N), and overnourished (O) [16]. All the states are transient or non-absorbing. The transitions are illustrated using diagrams with boxes and arrows. Boxes represent the states of CIAF, while arrows represent possible transitions between the states. As depicted in Fig 2, we have considered the 3 transient states and assumed that subjects would be in any given state at time t equal to zero (t = 0); then 6 possible transitions were identified by arrows. Transition rate denoted by

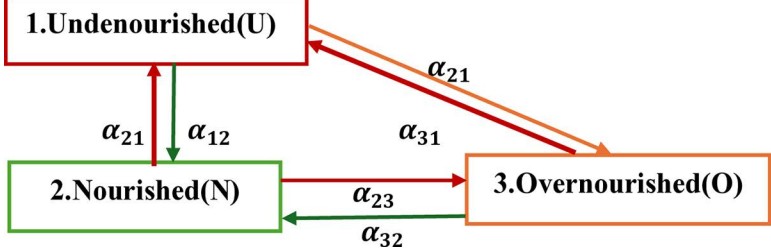

**Fig 2. Model of progression between CIAF states for U5C advances between adjacent or to non-adjacent states and optionally recovery to an adjacent/ non-adjacent state.**

$\alpha_{hj}$, *where, h and j are states, can take values :* $1, 2, 3$, that is transition from h$^{th}$ state to j$^{th}$ state with $\alpha_{hj}$ transition rate. For example: 1) from "U" to "N" by $\alpha_{12}$; 2) from "N" to "U" by $\alpha_{21}$; 3) from "N" to "O" by $\alpha_{23}$ 4) from "O to "N" by $\alpha_{32}$; 5) from "U" to "O" by $\alpha_{13}$ and 6) from "O to "U" by $\alpha_{31}$. The child in healthy state at time t can move either to under-nourished state or move to over–nourished state or stay at the same state after time t+1.

A multi-state process is a stochastic process (X $_{(t)}$, t∈T) with a finite state space $S = \{1,..., N\}$ *in* this case, S={1,2,3=N} since maximum number of states of CIAF is 3 and fulfilling some simplification assumptions. Here, T= [0, τ], τ < ∞ is the time interval and the value of the process at time t, is the state occupied at that time. This process has information about the different transitions that occur to an individual over time, as well as the time at which these transitions take place. In the process of nutritional status, the exact time of transition between states is unknown. The change of state can be observed after the follow-up time or the next round survey. Thus, the process could be continuous- time multi-state model or continuous-time Markov model.

The process starts with the distribution of the initial state probability given by: $P_j(0) = P[X(0) = j], j \in S$ and probabilities, $P_j(t) = P[X(t) = j], j \in S$ state occupation probabilities. With the evolution of the process over time, a history H$_t$ (a σ-algebra), would be generated consisting of the observation of the process over the interval [0, t), such as the states previously visited, times of transitions, etc. This multi-state process is fully characterized through transition probabilities and transition intensities. The transition probabilities between state h and state j at two times s and t can expressed by:

$$P_{hj}(s, t) = P(X(t) = j/X(s) = h, H_{s-}) \text{ for } h, j \in S, \; s, t \in T, \; s \leq t \tag{1}$$

This is the probability that a child in state *h,* at time *s* moves to state *j* by time *t*, conditional on the process history up until the time just before *s*, *Hs-*, where *h, j* ∈ *S*. This can be simplified with a Markov model, which assumes that the probability in (1) is only conditional on the state at time *s* and no other process history:

$$P_{hj}(S, t) = P(X(t) = j|X(s) = h, H_{s-}) = P(X(t) = j|X(s) = h) \tag{2}$$

The transition intensities, from state *h* to state *j* at time *t* is:

$$\alpha_{hj}(t) = \lim_{\Delta t \to 0} \frac{P_{hj}(t, \; t + \Delta t)}{\Delta t} = \frac{P(X(t + \Delta t) = j|X(t) = h)}{\Delta t} \tag{3}$$

This represents the instantaneous rate, at which a child leaves a nourished state, h and enters an undernourished or overnourished state j at time t. Let $Q(t)$ denote the transition matrix with (h, j) entry α$_{hj}$ (t) at time t for U5C and defined for 3 states as:

$$Q(t) = \begin{pmatrix} \alpha_{11} & \alpha_{12} & \alpha_{13} \\ \alpha_{21} & \alpha_{22} & \alpha_{23} \\ \alpha_{31} & \alpha_{32} & \alpha_{33} \end{pmatrix} \tag{4}$$

Where, the diagonal element of the row is equal to negative of the sum of other entries of that row.

The estimation of MSM is started with, is the probability of being in any given state at a given time *t*. It can be estimated using the non-parametric Aalen-Johansen estimator and plotted over time in a similar manner to the Kaplan–Meier curves. The curves show the likelihood of a subject being in one of the states over time, [24]. The transition probability matrix helps to estimate how probable children's nutritional status is in each state as time changes. That is how children are probable to move between states of CIAF as time passes. Let P(t) denote the transition probability matrix where each element $\pi_{hj}(t)$ represents the probability that a child transitions from state h to state j over the time interval [s, t], given that

the child was in state h at time s. Thus, the matrix $P(t)$ is typically derived from the transition intensity matrix $Q(t)$ under the assumption of a time-homogeneous continuous-time Markov process, [26] such that:

$$P(t) = \exp[Q(t)] = \begin{pmatrix} \pi_{11} & \pi_{12} & \pi_{13} \\ \pi_{21} & \pi_{22} & \pi_{23} \\ \pi_{31} & \pi_{32} & \pi_{33} \end{pmatrix}$$

(5)

The transition intensities can be estimated by maximum likelihood procedures as a product of probabilities of transition between observed states, overall individuals $i = 1, 2,.., M$ and observation times $r$ which are observed $n$ times, as shown below:

$$L(Q) = \prod_{i=1}^{M} \prod_{r=1}^{n_i-1} L_{ir} = \prod_{i,r} \pi_{s(t_{ir})s(t_{i,r+1})}(t_{i,r+1} - t_{ir})$$

(6)

Mean sojourn time or expected time of stay in a given state before transition and total time of stay in a given state during whole process of a child can be computed by [27]:

$$mean\ sojourn\ time = \frac{-1}{\alpha_{rr}},\ where,\ \alpha_{rr}\ diagonal\ entry\ of\ transition\ matrix$$
$$total\ time = \int_{t_1}^{t_2} P_{ik}(t)dt,\ where,\ t_1\ is\ initial\ entry\ time\ of\ a\ given\ state\ and\ t_2\ is\ final\ time\ of\ the\ process\ ,$$

**2.3.2. Multi-state regression models.** To identify the significant factors of transition, it is important to relate the individual characteristics with the intensity rates through a covariate vector, $Z$, possibly time dependent. The inference in a MSM can be divided into several survival models, by fitting separate intensities to all possible transitions. For a general regression model of hazard or failure rate for $i^{th}$ child moving from state h to j is a function of time t and covariate vector Z. It could be expressed as:
$\alpha_{hji}(t,\ Z) = \phi\left(\alpha_{hj0}(t),\ \beta_{hj}^T Z_i\right)$, where $\alpha_{hj0}(t)$ is the baseline intensity function, $\beta_{hj}$ is the vector of regression coefficient, and $Z_i$ is the covariate vector for subject $i$. A popular choice that simplifies the model for inference is the proportional hazards assumption, which is obtained by:

$$\phi(u(t),\ v) = u(t)\ e^v\ implies, \alpha_{hj}(t,z) = \alpha_{hj0}(t)\ \exp\left(\beta_{hj}^T Z_i\right)$$

(7)

A Cox proportional hazards model of type

$$\alpha_{hj}(z) = \alpha_{hj}\exp\left(\beta_{hj}^T Z\right),\ Where,\ 1 \le h < j \le 3,$$

(8)

Where, 3 CIAF states were considered to relate the transition intensities $\alpha_{hj}$ with child's characteristics or covariates, $Z$. The inclusion of covariates in these models allows the prediction of probabilities tailored to individual child. The inference is based on the ML method in (6) by replacing the transition intensities $\alpha_{hj}$ by those given above in (8), [28].

**2.3.3. Software.** Data clearance and analysis were performed by SPSS version 27, and R version 4.3.2 with msm 1.7.1. The standard scores of all anthropometric measure indexes were computed by zscorer 0.3.1 packages. The R codes were adapted from manual of msm 1.7.1 Package by Jackson 2023.

## 3. Result interpretations

### 3.1. Descriptive result

As presented in S1 Table, the prevalence of malnutrition at each round was identified. The prevalence of under-five stunting was 1140 (37.5%), 1560 (52.5%), and 1292 (46%) at the baseline, midline, and endline rounds, respectively.

Prevalence of wasting at the baseline, midline, and endline rounds was 409 (13.5%), 185 (6.2%), and 240 (8.7%), respectively. In addition, the underweight prevalence at baseline, midline, and endline was 723 (23.8%), 811 (27.3%), and 835 (29.7%), respectively. Table 2 shows that in the baseline, midline, and endline rounds, 3,044, 2,973, and 2,813 children were measured at least twice, respectively. The prevalence of undernutrition at baseline, midline, and endline rounds, respectively, was 1207 (39.7%), 1481 (49.8%), and 1334 (47.4%), while the prevalence of overnutrition was 423 (13.9%), 269 (9%), and 219 (7.8%), at baseline, midline, and endline rounds respectively.

The descriptive result in Table 3 reveals that 85.9% of mothers or primary caregivers were married and living with a single spouse. A significant proportion (72.8%) of them had no formal education, and 67.9% of them were identified as housewives. Additionally, 84.4% reported that their spouses or partners were living with them at the time of the first-round survey. Among the U5C included in the study, 1569 (51.5%) are males. Of the households selected for the baseline survey, 1707 (56.1%) were from the Amhara region, specifically from the North Wollo (36.9%) and Waghimra (19.2%) zones. The remaining 1337 households (43.9%) were from the Oromia region, particularly from the East Hararge (9%), West Hararge (32.3%), and West Arsi (2.6%) zones. Majorities of households (56.5%) included in the study were Orthodox religion followers, followed by Muslim (41.7%). Slight changes in these proportions were observed across subsequent survey rounds, which may be attributed to loss to follow-up and other longitudinal reasons.

## 3.2. Multistate model result

Fig 3 illustrates the transitions between three states of CIAF over the study period. The result shows that children's nutritional status changed over time, transitioning between the healthy, undernourished, and overnourished states. For example, the child represented by the black line was in an undernourished state at the baseline, recovered to a healthy state by 18 months, and remained healthy after the midline survey. In contrast, the yellow line represents a child who remained in the undernourished state throughout the entire study period. The gray line depicts a child who started in a healthy state at baseline, transitioned to an undernourished state at 18 months, and then moved to an overnourished state during the endline survey. Meanwhile, the pink line shows a child who was in an overnourished state at baseline, transitioned to an undernourished state at 18 months, and had no follow-up data after the midline survey. These transitions suggest that movement between all three states is possible and that individual children may improve, deteriorate, or remain in the same nutritional state throughout the study period. The figure highlights the dynamic nature of nutritional status among under five children in a longitudinal setting.

As shown in Table 4, among the U5C included in the study, 698 (27.4%) of them transitioned from a healthy state to an undernourished state, while 173 (6.8%) transitioned to an overnourished state. These indicate that the number of U5Cs transitioning from a healthy state to an undernourished state was approximately four times higher than the number of children transitioning from a healthy state to an overnourished state. Among those in an undernourished state, 545 (21.3%) recovered back to a healthy state. Similarly, 260 children (38.5%) from an overnourished state recovered to a healthy state, indicating a relatively higher recovery rate in the overnourished group. Additionally, a significant proportion

**Table 2. Composite Index Anthropometric Failure (CIAF) at the baseline, midline and endline rounds.**

| Response | States | Rounds | | | | | |
|---|---|---|---|---|---|---|---|
| | | Baseline | | Midline | | Endline | |
| | | Freq | % | Freq | % | Freq | % |
| Composite index anthropometric status of U5C | U (1) | 1207 | 39.7% | 1481 | 49.8% | 1334 | 47.4% |
| | N (2) | 1414 | 46.5% | 1223 | 41.1% | 1260 | 44.8% |
| | O (3) | 423 | 13.9% | 269 | 9.0% | 219 | 7.8% |
| | Total | 3044 | 100.0% | 2973 | 100.0% | 2813 | 100.0% |

**Table 3. Mother, Child and Household Characteristics.**

| Variables | Categories | Rounds | | | | | |
|---|---|---|---|---|---|---|---|
| | | Baseline | | Midline | | Endline | |
| | | Freq | % | Freq | % | Freq | % |
| Marital status of the biological mother or the primary caregiver | Married, Single Spouse | 2613 | 85.9% | 2534 | 85.2% | 2424 | 86.2% |
| | Married, More than one spouse | 10 | 0.3% | 18 | 0.6% | 1 | 0.0% |
| | Not together for any reason | 92 | 3.0% | 57 | 2.0% | 40 | 1.4% |
| | Divorced | 228 | 7.5% | 250 | 8.4% | 232 | 8.3% |
| | Widowed | 101 | 3.3% | 114 | 3.8% | 116 | 4.1% |
| Education Level of biological mother or primary caregiver | Did not complete any schooling | 2215 | 72.8% | 2081 | 70.0% | 1968 | 70.0% |
| | Adult, religious or other | 48 | 1.6% | 68 | 2.3% | 56 | 2.0% |
| | Primary School | 707 | 23.2% | 737 | 24.8% | 704 | 25.0% |
| | High School and above | 74 | 2.4% | 87 | 2.9% | 85 | 3.0% |
| The main current activity of the biological mother or the primary caregiver | Crop or livestock production | 214 | 7.0% | 877 | 29.5% | 787 | 28.0% |
| | business or skilled labor | 673 | 22.1% | 274 | 9.2% | 259 | 9.2% |
| | Employee | 2 | 0.1% | 12 | 0.4% | 13 | 0.4% |
| | Unpaid house work | 2066 | 67.9% | 1767 | 59.4% | 1710 | 60.8% |
| | Student, volunteer, and other | 89 | 2.9% | 43 | 1.5% | 44 | 1.6% |
| Spouse or partner is a member of the household? | Yes | 2570 | 84.4% | 2374 | 79.9% | 2323 | 82.6% |
| | No | 474 | 15.6% | 599 | 20.1% | 490 | 17.4% |
| Sex of Baseline Indexed Child | Male | 1569 | 51.5% | 1534 | 51.6% | 1456 | 51.8% |
| | Female | 1475 | 48.5% | 1439 | 48.4% | 1357 | 48.2% |
| Region | Amhara | 1707 | 56.1% | 1662 | 55.9% | 1578 | 56.1% |
| | Oromia | 1337 | 43.9% | 1311 | 44.1% | 1235 | 43.9% |
| Zone | North Wollo | 1123 | 36.9% | 1100 | 37.0% | 1055 | 37.5% |
| | Waghimra | 584 | 19.2% | 562 | 18.9% | 523 | 18.6% |
| | East Hararge | 273 | 9.0% | 267 | 9.0% | 261 | 9.3% |
| | West Arsi | 80 | 2.6% | 72 | 2.4% | 76 | 2.7% |
| | West Hararge | 984 | 32.3% | 972 | 32.7% | 898 | 31.9% |
| Religion of household head | Orthodox | 1721 | 56.5% | 1673 | 56.3% | 1583 | 56.3% |
| | Muslim | 1270 | 41.7% | 1252 | 42.1% | 1175 | 41.8% |
| | Protestant | 53 | 1.8% | 48 | 1.6% | 55 | 1.9% |
| Main language of household head | Agew (Hemrigna/Awigna) | 244 | 8.0% | 165 | 5.5% | 144 | 5.1% |
| | Amharic | 1466 | 48.2% | 1510 | 50.8% | 1437 | 51.1% |
| | Oromigna | 1303 | 42.8% | 1289 | 43.4% | 1222 | 43.4% |
| | Other Ethiopian Languages | 31 | 1.0% | 9 | 0.3% | 10 | 0.4% |
| Sex of household head | Male | 2486 | 81.7% | 2440 | 82.1% | 2305 | 81.9% |
| | Female | 558 | 18.3% | 533 | 17.9% | 508 | 18.1% |
| Marital status of household head | Married, Single Spouse | 2562 | 84.2% | 2458 | 82.7% | 2411 | 85.7% |
| | Married, more than one spouse | 20 | 0.7% | 72 | 2.4% | 0 | 0.0% |
| | Not together for any reason | 75 | 2.5% | 49 | 1.6% | 33 | 1.2% |
| | Divorced | 240 | 7.9% | 249 | 8.4% | 229 | 8.1% |
| | Widowed | 147 | 4.8% | 145 | 4.9% | 140 | 5.0% |
| Highest class household head attended | Did not complete any class | 2030 | 66.7% | 1663 | 55.9% | 1572 | 55.9% |
| | Adult/religious &other school | 119 | 3.9% | 192 | 6.5% | 183 | 6.5% |
| | Primary School | 796 | 26.1% | 1000 | 33.6% | 945 | 33.6% |
| | High School and above | 99 | 3.3% | 118 | 4.0% | 113 | 4.0% |

*(Continued)*

**Table 3.** (Continued)

| Variables | Categories | Rounds | | | | | |
|---|---|---|---|---|---|---|---|
| | | Baseline | | Midline | | Endline | |
| | | Freq | % | Freq | % | Freq | % |
| The main current activity of household | Crop or livestock production | 913 | 30.0% | 2443 | 82.2% | 2392 | 85.0% |
| | business or skilled labor | 1774 | 58.3% | 248 | 8.3% | 225 | 8.0% |
| | Employee | 12 | 0.4% | 19 | 0.6% | 15 | 0.5% |
| | Unpaid house work | 223 | 7.3% | 177 | 6.0% | 118 | 4.2% |
| | Student, volunteer extra | 122 | 4.0% | 84 | 2.8% | 63 | 2.2% |

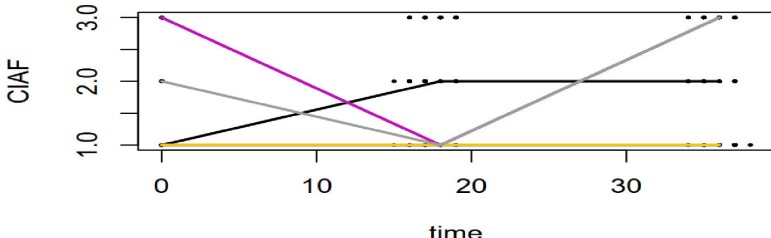

**Fig 3. Spaghettis plot of CIAF for 4 unique ID selected individual children from data.**

**Table 4. State Table or Frequency of Transition at End of Study.**

| | To | | | Total |
|---|---|---|---|---|
| From | Under Failure (UF) | Normal Only (NO) | Over Failure (OF) | |
| UF (1) | 1838(71.82%) | 545(21.3%) | 176(6.88%) | 2559 |
| NO (2) | 698(27.4%) | 1678(65.8%) | 173(6.8%) | 2549 |
| OF (3) | 279(41.27%) | 260 (38.46%) | 137 (20.27%) | 676 |
| Total | 2815 | 2483 | 486 | 5784 |

of children remained in their current health states throughout the study period. Specifically, 1,838 children (71.82%) remained in a healthy state, 1,678 children (65.8%) remained in an undernourished state, and 137 children (20.27%) remained in the overnourished state. The aim of the study was to understand and quantify the transition rates of U5C between CIAF states. The finding suggests that while most children maintain their health status, a notable proportion experience transitions, with a higher tendency to move into the undernourished state compared to the overnourished state. The recovery rates indicate that a substantial proportion of children in both failure states manage to return to a healthy state during the study period.

The result in Fig 4 compares the expected (red dotted line) and observed (blue line) prevalence in each nutritional state. The closeness of lines across all states suggests that the continuous-time Markov multi-state model best fits the data. In state 3, the breakpoint shows that there is a noticeable break in the data after 15 months. These indicate a lack of sufficient data for both the expected and observed percentages. The possible reason could be the households included in the study were vulnerable to food insecurity. Therefore, children are at a higher risk of undernutrition and are not stable in an overnutrition state for an extended period. This result supports the fact that the model fits well in general.

Table 5 shows the baseline hazard and model information criterion. The −2LL and AIC values are smaller for the model with covariates compared to the model without covariates, suggesting that the model with covariates provides a better fit

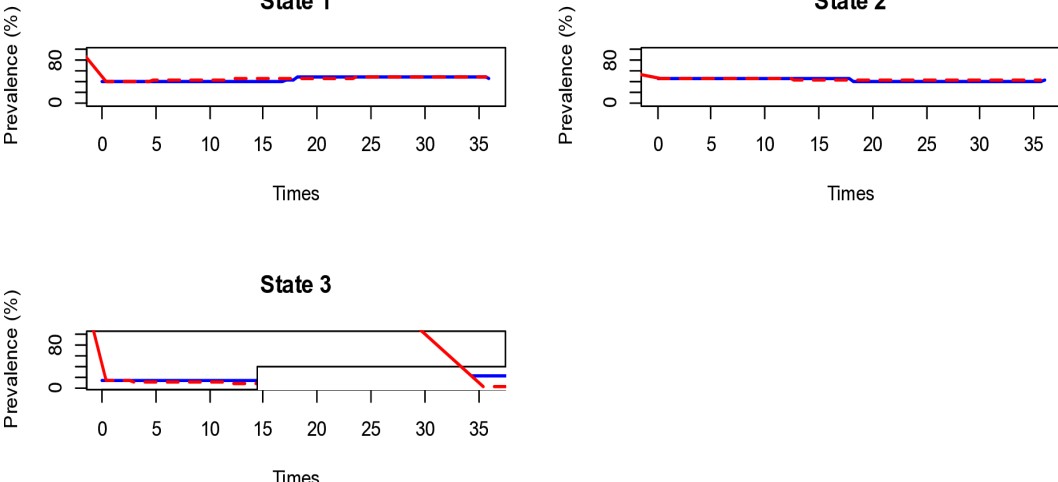

**Fig 4. Prevalence plot, which indicates percentage of estimated and observed prevalence.**

**Table 5. Estimated Values of Baseline Transition Intensity and Corresponding 95% Confidence Interval from the Multi-state Model with and without Covariates.**

| Transition Intensities without Covariate | | | Transition Intensities with Covariate | |
|---|---|---|---|---|
| **From State to state** | **Baseline MLE** | **95% CI** | **Baseline MLE** | **95% CI** |
| State 1 – State 1 | −0.019736 | (−0.0216, −0.018) | −0.024131 | (−0.0263, −0.0222) |
| State 1 – State 2 | 0.0126 | (0.0111,0.0142) | 0.014625 | (0.0127, 0.0168) |
| State 1 – State 3 | 0.007183 | (0.0057, 0.0091) | 0.008735 | (0.0070, 0.0109) |
| State 2 – State 1 | 0.022356 | (0.0203, 0.0247) | 0.020298 | (0.0183, 0.0226) |
| State 2 – State 2 | −0.030511 | (−0.033, −0.0283) | −0.028542 | (−0.0311, −0.0262) |
| State 2- State 3 | 0.008154 | (0.0063, 0.0105) | 0.008244 | (0.0064, 0.0106) |
| State 3 – State 1 | 0.063464 | (0.0522, 0.0772) | 0.046233 | (0.0379, 0.0564) |
| State 3 – State 2 | 0.048562 | (0.0386, 0.0612) | 0.063825 | (0.0511, 0.0797) |
| State 3 – State 3 | −0.112026 | (−0.1292, −0.097) | −0.103829 | (−0.117, −0.0922) |
| −2 * log-likelihood | 9419.898 | | 9194 | |
| AIC | 9431.898 | | 9230.347 | |
| Df | 6 | | 18 | |

to the data. Therefore, the interpretation is based on the model that includes covariates. Estimating the transition rates or hazard rates to either failure states or recovery from them is the primary aim of the study. Thus, the estimated result reveals that the baseline transition intensities of the nourished (state 2) to undernourished and overnourished states were 0.020298 and 0.008244, respectively. These indicate that children in a nourished state were 2.45 (0.020298/0.008244) times more likely to transit to an undernourished state (state 1) than to an overnourished state (state 3). Children in an overnourished state were 4.34 (0.063825/0.014625) times more likely to recover back to a nourished state compared to children in an undernourished state. This shows that children in the nourished state had a higher risk of transiting to an undernourished state, but a lower likelihood of recovering compared to children in an overnourished state.

Table 6 depicts the estimated 1-month and 36-month probabilities of transition, mean sojourn time, and total time of subject spent in a given state. The 1-month probabilities of transition show that U5Cs staying in the current states are more probable. That means the likelihood of staying in an undernourished state, nourished state, and overnourished state

**Table 6. Estimated 1-month and 36-moths Probabilities of Transitioning between CIAF States, the Mean Sojourn and Total time of stay.**

| CIAF Category | Transition Probability Matrix (t=1- month)1-month | | | Mean Sojourn Time |
|---|---|---|---|---|
| | Undernourished (1) | Normal (2) | Overnourished (3) | MST (95%CI) |
| | MLE (95%CI) | MLE (95%CI) | MLE (95%CI) | |
| Undernourished (1) | 0.976254 (0.97419,0.97808) | 0.015220 (0.01352,0.01695) | 0.008526 (0.00682,0.01035) | 40.96 (37.6, 44.62) |
| Nourished (2) | 0.020532 (0.01859,0.02251) | 0.971391 (0.96911,0.97351) | 0.008077 (0.00637,0.00995) | 34.01 (31.45, 36.78) |
| Overnourished (3) | 0.047796 (0.04001,0.05639) | 0.049078 (0.04103,0.05788) | 0.903125 (0.89134,0.91303) | 9.77 (8.69, 10.99) |
| **Transition Probability Matrix (t=36 months)** | | | | **Total time of stay** |
| CIAF Category | Undernourished (1) | Normal (2) | Overnourished (3) | Total time (95%CI) |
| Undernourished (1) | 0.6020467 | 0.3196602 | 0.07829304 | 26.79802 (26.27, 27.32) |
| Nourished (2) | 0.4052930 | 0.5177302 | 0.07697676 | 7.022167 (6.51, 7.49) |
| Overnourished (3) | 0.4852087 | 0.4189034 | 0.09588796 | 2.179810 (1.89,2.48) |

was 97.63%, 97.14%, and 90.3%, respectively. Children in a nourished state were 1.35 (0.020532/0.01522) times more likely to transit to an undernourished state than to backward transition. Moreover, the transition from an overnourished state to an undernourished state is 5.6 (0.0478/0.00853) times more probable than that of backward transition. A child is sustained in a nourished state for 34 months on average. However, a child spent 41 months in an undernourished state before transitioning to any other state. A typical child in a healthy state has a 40.53% probability of becoming undernourished after 3 years. That is, the likelihood of a child moving from a nourished state to an undernourished state is 1.3 times higher than a backward transition. However, probabilities of children remaining in the same states, such as undernourished, nourished, and overnourished, in the duration of 36 months are 0.6, 0.52, and 0.1, respectively. The total time of staying in an undernourished state, nourished state, and overnourished state was 27 months with 95% CI: (26.27, 27.32), 7 months with 95% CI: (6.51, 7.49), and 2.2 months with 95% CI: (1.89, 2.48), respectively, per given period. It shows that the lifespan of U5Cs spent in undernourished, nourished, and overnourished states was 75%, 20%, and 5%, respectively. These findings highlight the predominance of undernourishment among U5C in the study area, with children spending most of their early years in this compromised health state. The finding also suggests that children tend to remain in the undernourished state for an extended period before transitioning to either nourished or overnourished states. Such patterns are indicative of this study's aim being impactful in the manner of a multi-state model, utilized in nutrition to represent different statuses and the transitions between them. In this model, each state (nourished, undernourished, or overnourished) is associated with specific transition probabilities, and the mean sojourn time reflects the expected duration an individual stays in a particular state before moving to another.

Table 7 reveals the probability of being in the next state. Its confidence interval was estimated by bootstrapping with boot number 1000. It was known that children were to be moved from a nourished state, and then they had a 71% chance of becoming undernourished in the next. Similarly, children in an overnourished state had a 49.2% risk of being in an undernourished state in the next. However, the children in an undernourished state had a 63.1% chance of recovering, while there is a 36.9% chance of becoming overnourished in the next.

**Table 7. Probability of Children in Each state Being Next.**

| | Undernourished (1) | Nourished (2) | Overnourished(3) |
|---|---|---|---|
| State 1 | 0 | 0.631 (0.5608,0.6920) | 0.369 (0.3080,0.4392) |
| State 2 | 0.71 (0.6560,0.7633) | 0 | 0.2900 (0.2367,0.3440) |
| State 3 | 0.4919 (0.4254,0.5576) | (0.4424,0.5746) | 0 |

Table 8 shows the effects of explanatory variables on transition intensities. Girls were 0.8013 times less likely to transition to an undernourished state (HR, 0.8013, 95% CI (0.6730, 0.9542)) and had 1.824 times higher likelihood of recovering from an overnourished state (HR, 1.824, 95% CI (1.3439, 2.475)) than boys. The likelihoods of transitioning from a nourished state to an undernourished and overnourished state were reduced by 0.031 and 0.034 times, respectively, as the age increased by one month. One month increase in the age of children who were in the undernourished and overnourished states was associated with higher recovery chances (HR = 1.013, 95% CI: 1.005, 1.022) and (HR = 1.036, 95% CI: 1.023, 1.050), respectively. Children of educated mothers were less likely to transition from a nourished state to an undernourished state compared to children whose mothers had no formal education (HR: 0.8171, 95% CI (0.6685, 0.9987)). Children from the Oromia region were (HR: 0.8074, 95% CI (0.6681, 0.9756)) and (HR: 0.4290 (0.2784, 0.6612)) times less likely to transition to undernourished and overnourished states, respectively, than children in the Amhara region. This result emphasizes meeting the objective of identifying the driving factors of transition. It suggests the importance of gender, maternal education, age, and regional factors in the transition between different health states in children.

## 4. Discussion

This study sought to investigate malnutrition of U5C in the Amhara and Oromia regions. The study introduced CIAF as a response variable by aggregating nine different combinations of malnutrition categories, such as stunting, wasting, underweight, and overweight. Applying the CIAF contributes to having a single conclusion for overall malnutrition. In addition, the study also tried to incorporate the dynamic nature of children's malnutrition. The continuous-time multi-state Markov model was applied to estimate transition dynamics and their associated factors between states. Then, the key findings were discussed in three themes as follows:

### 4.1. Prevalence of malnutrition

The prevalence of stunting was consistently high, peaking at midline and showing some improvement by the endline. That is, it increased from 37.5% at baseline to 52.5% at midline but decreased slightly to 46% at endline. This might suggest that interventions or changes between the midline and endline might have contributed to reducing the prevalence of stunting, but the overall prevalence remained concerning. Wasting decreased from 13.5% at baseline to 6.2% at midline but slightly increased again to 8.7% at endline. It showed a positive trend at the midline but worsened slightly at the endline, signaling the need for continuous interventions to address this issue. However, the underweight trend worsened consistently, which might indicate broader, more systemic issues affecting nutrition, growth, or health outcomes that are not being sufficiently addressed. It showed a steady increase, from 23.8% at baseline to 29.7% at endline, with a peak at

**Table 8. Hazard Ratios with 95% Confidence Intervals for Transition Rates across CIAF States by Covariates.**

| Covariates | From nourished CIAF to Undernourished | From Undernourished to nourished CIAF | From nourished CIAF to Over-nourished | From Overnourished to nourished CIAF |
|---|---|---|---|---|
| | HR (95%CI) | HR (95%CI) | HR (95%CI) | HR (95%CI) |
| Sex (Male Ref) | 0.8013 (0.6730,0.9542) | 0.918 (0.7346,1.147) | 0.7360 (0.5019,1.0794) | 1.824 (1.3439,2.475) |
| Age in months | 0.9693 (0.9634,0.9752) | 1.013 (1.005,1.022) | 0.9662 (0.9538,0.9788) | 1.036 (1.023,1.050) |
| Mother Occupation (homeworker ref) | 0.8506 (0.7060,1.025) | 0.9838 (0.7824,1.237) | 1.2580 (0.8606,1.839) | 1.0105 (0.7467,1.367) |
| Mother education (did not complete any school ref) | 0.8171 (0.6685,0.9987) | 1.196 (0.9367,1.528) | 0.9332 (0.6103,1.4267 | 1.094 (0.7868,1.520) |
| Household size (five or less members ref) | 0.9711 (0.7999,1.179) | 0.9908 (0.7852,1.250) | 1.2845 (0.8444,1.954) | 0.8584 (0.6213,1.186) |
| Region (Amhara ref) | 0.8074 (0.6681,0.9756) | 1.114 (0.8878,1.399) | 0.4290 (0.2784,0.6612) | 1.214 (0.8781,1.677) |

27.3% at midline. Furthermore, the prevalence of undernourishment increased from 39.7% at baseline to 49.8% at midline, then dropped slightly to 47.4% at endline. Undernourished state followed a similar trajectory as stunting and underweight, but the trend did show a slight improvement during the endline. Conversely, the overnourished rate decreased from 13.9% at baseline to 7.8% at endline, with a decrease to 9% at midline. In general, the malnutrition indexes were higher than recent national estimates. The national prevalence estimates were 34.4% for stunting, 6.8% for wasting, and 21% for underweight [5,7].

## 4.2. Finding of duration in malnutrition states

In the study area, the majority (65.8%) of U5C from PSNP member households remained in an undernourished state. Among the nourished children, the likelihood of transitioning to an undernourished state was four times higher than transitioning to an overnourished state. Similarly, the proportion of children moving from an overnourished to an undernourished state was 1.1 times higher than those transitioning to a healthy state. These findings are consistent with those of [19], but contrast with the results reported by [25,29,30]. The current study and the study by Oduro et al. were conducted in neighboring countries, Ethiopia and Kenya. Therefore, the consistency in findings may be due to similarities in socioeconomic status, dietary habits, and healthcare infrastructure in these countries. In contrast, the studies by Häkkänen et al., Hu et al., and Moreira et al. were conducted in developed countries, where overnutrition is the predominant form of malnutrition. So, the underlying causes, risk factors, and public health responses to malnutrition may differ significantly between these settings. Furthermore, children in the overnourished state were 1.81 times more likely to recover to normal nutritional status compared to those in the undernourished state. This may be attributed to the fact that children from the poorest households are more vulnerable to undernutrition due to limited access to adequate food and healthcare. Additionally, children in the overnourished state were more likely to transition to an undernourished state than to return directly to a nourished state, indicating instability in their nutritional condition.

The estimated 1-month transition probabilities show that the chance of staying in the current state is higher than transitioning to other states. This result is supported by previous studies in Finland, China, Portugal, and Kenya [19,25,29,30]. That is, the highest (98%) 1-month chance of staying in the current state was observed in the undernourished state. In contrast to this, the highest chance was observed in an overweight state in [25,29,30] and at nourished state in [19]. The likelihood of recovery from an undernourished state was 3.22 times lower than the likelihood of transitioning into it. Additionally, the risk of transitioning from an overnourished state to an undernourished state was 4.7 times higher than the reverse transition. This could be due to food insecurity among the households surveyed, which may have prevented children from maintaining an overnourished state. In a given time, children in either a nourished or an overnourished state faced a higher risk of falling into an undernourished state than transitioning to any other nutritional category. Once U5Cs become undernourished, it takes an average of 3.5 years to recover to normal nutritional status. Conversely, a child remains in the nourished state for approximately 34 months before transitioning to malnutrition. In contrast, [29] found that the longest duration was spent in the obesity state, while [30] reported the longest duration in the nourished state. In our findings, a child in a nourished state had a 71% higher risk of transitioning to an undernourished state than to an overnourished state in the next. Moreover, U5Cs spent approximately 75% of their lifetimes in an undernourished state. This differs from previous studies, where longer durations were observed in the obese or nourished states [29,30]. These differences may be explained by contextual factors at the time of the surveys, such as ongoing civil conflicts, locust infestations, and recurring droughts in the study areas. These combined shocks likely contributed to prolonged periods of undernutrition among U5Cs in the study area.

## 4.3. Finding of covariates effects

**Gender Differences:** Girls were less likely to transition into an undernourished state and more likely to recover from the overnourished state compared to boys. This finding is consistent with previous research [11,19,25,30–32]. Cultural effects

on gender perceptions may partly explain these differences. In the Ethiopian context, especially in rural communities, boys are often prioritized for what are considered "superior" feeding options. As a result, boys tend to receive complementary foods earlier than girls because breast milk is sometimes viewed as inferior. This early introduction of complementary foods—particularly in settings with poor hygiene—can increase the risk of infections. In addition, boys are often allowed more freedom to play outdoors, increasing their exposure to environmental pathogens. This greater mobility, while culturally supported, may raise the likelihood of infections that impair nutrient absorption and worsen malnutrition. Biological differences may also play a role. Research shows that male placentas are relatively smaller in proportion to birth weight compared to female ones, potentially limiting reserve capacity during periods of nutritional stress [33,34]. This makes male infants more vulnerable to food shortages. Additionally, girls may have stronger immune responses, particularly in their ability to produce antibodies, which offers them better protection against infection-related malnutrition [35]. The issue is more pronounced among households in lower socioeconomic groups and those facing higher levels of food insecurity. Older child age is positively associated with healthier nutritional status. This finding is consistent with previous studies [11,13,17,19,25,29,31,36,37].

**Impact of Maternal Education:** Children of mothers with formal education are less likely to transition from a healthy nutritional state to an undernourished state compared to children whose mothers had no formal education. This finding aligns with the results of previous studies [10,11,17–19,25,31,32,36–38]. This may be because educated mothers are generally more aware of child nutrition, diet diversity, hygiene practices, and appropriate childcare. As a result, they are better equipped to prevent malnutrition and promote healthy growth in their children.

**Regional Differences:** Children living in the Oromia region had less risk of transitioning from a healthy nutritional state to malnutrition compared to children in the Amhara region. This finding is consistent with previous studies [10,11,13,18]. The observed difference may be attributed to several contextual factors affecting the Amhara region during the data collection period. These include the civil conflict in northern Ethiopia, widespread locust infestations, and recurring droughts. The zones selected from Amhara were particularly affected by these overlapping shocks, which likely had a more severe impact on child nutritional status in Amhara compared to Oromia.

## 4.4. Strength and limitation of the study

The strengths of this study include the utilization of a larger dataset collected by an international organization. The sample includes rural households affected by food insecurity, which highlights areas in need of targeted interventions based on the study findings. The study utilized the CIAF as an indicator of child malnutrition, aggregating conventional indexes to provide a more comprehensive conclusion. In addition, the response exhibits a transient or dynamic nature. To estimate the varying risks of transition and their associated covariate effects, the study employed a continuous-time multistate Markov model (CTMSM). The application of a CTMSM in studying child nutritional dynamics is not only methodologically novel but also offers significant practical implications for public health planning and policy. First, unlike traditional models, in CTMSMs, the transition between nutritional states at any point in continuous time aligns with the real-world nature of child growth and health deterioration, which do not occur at fixed intervals. This allows for more accurate modeling of when and how children are likely to shift between nutritional states. Second, the model handles intermediate, multiple, and recurrent transitions, making it particularly useful in capturing the complexity of nutritional status over time. It enables practitioners to distinguish between temporary deterioration and persistent malnutrition. Third, the use of CTMSMs in settings affected by conflict, drought, or food insecurity (such as Ethiopia) provides a more nuanced understanding of how such environmental shocks influence both the direction and speed of nutritional transitions. Importantly, CTMSMs provide quantitative estimates such as transition probabilities, sojourn times, and state occupation probabilities, which can be used to forecast future malnutrition trends under different scenarios. For instance, health planners can simulate how nutritional outcomes might evolve under improved food access or the continuation of environmental shocks like drought. This makes the model a valuable decision-support tool for long-term planning. Furthermore, by identifying states

where children are most likely to remain for prolonged periods (e.g., undernourished states), the model can help prioritize resource allocation to high-risk groups or regions. It can guide where to focus limited nutrition resources, such as supplementary feeding programs or health education campaigns, and when to intervene for maximum impact. While CTMSMs have been widely used in areas like oncology and infectious disease progression, their application in child malnutrition research is still emerging. Applying CTCSMs in the field of nutrition represents a methodological innovation; it could offer a robust, flexible framework for informing evidence-based policies and interventions in complex, resource-limited settings and dynamic nutrition issues. However, this study could not be considered conclusive for all households at the country level or even across all study areas. The sample was limited to households participating in the social safety net program, so the results may not fully represent the entire population. Nevertheless, this study serves as a useful starting point for national surveys and related studies at the country level, particularly for applying the CIAF and multi-state models. Due to convergence issues, only a limited number of explanatory variables were included in the analysis.

### 4.4. Policy Implications

The current study revealed that U5Cs in the study area had an increased risk of transitioning to an undernourished state than transitioning to any other state. In addition, three-fourths of observed children's lifespans were spent in an undernourished state before transitioning to other states (Tables 5–6, and 7). Furthermore, being male, children under two years of age, mothers who had no education, and living in the Amhara region, particularly in the Waghimra and North Wollo zones, were significantly aggravating the risk of transitioning into malnutrition among U5C in the study area (Table 8). These results call for planning for policies of intervention on nutrition and dietary improvement strategies like poultry rearing, adopting and scaling up nutrition-sensitive agriculture, increasing productivity in the study areas, and setting other poverty reduction activities to improve nutrition and increase diet diversity in study areas beside PSNP [20] and planning for policies for adult education and training campaigns to boost knowledge and awareness of parents on cultural views of gender, childcare, and dietary diversity, especially fasting time among rural communities in the two regions and beyond. Unless interventions are implemented, it has a high impact on children's health and development, like irreversible physical and cognitive damage and recent rapid weight loss-related risk of death [1,2,4].

## 5. Conclusions and recommendations

### 5.1. Conclusions

The prevalence of undernourishment was above the national average. It increased from 39.7% at baseline to 47.4% at endline. The findings of this study reveal that transitions in the CIAF among U5C are bidirectional, indicating that children can move between different nutritional states over time. However, children in the study areas demonstrated a higher likelihood of transitioning into undernutrition than recovering back to a nourished state. It was evidenced by the fact that children who were initially in a nourished state had the highest probability of becoming undernourished in the subsequent period. Moreover, three-fourths of the lifespan of children was spent in an undernourished state. The study also identified several risk and protective factors that influence the likelihood of both forward (deterioration) and backward (recovery) transitions in children's CIAF status. Factors associated with an increased likelihood of transitioning to a nourished state include being female, being older than two years, and having a mother with formal education. These findings emphasize the importance of demographic and socioeconomic factors in shaping children's nutritional trajectories.

### 5.2. Recommendations

Based on the current findings, the identified risk factors need broad, targeted interventions and policies by governmental and non-governmental organizations (GO/NGOs) in rural communities. GO/NGOs should develop special intervention strategies to mitigate the impacts of recurrent shocks such as droughts, locust infestations, and conflicts, aiming to

shorten the duration children remain in a state of undernourishment. Furthermore, GO/NGOs or relevant stakeholders could implement targeted literacy campaigns, either community-based group sessions or mobile health interventions, that focus on enhancing maternal knowledge and skills related to improving child nutrition, dietary diversity, childcare, hygiene, and family planning. These campaigns should also aim to address and transform cultural beliefs and practices related to feeding habits, especially during fasting periods, and to challenge harmful cultural attitudes surrounding gender roles and their impact on family health and well-being. Utilizing health extension workers or other trained health and nutrition professionals can help effectively deliver these messages at the community level. In the long run, the Ministry of Health, the Ministry of Agriculture, and other relevant stakeholders should strengthen efforts to reduce poverty and improve the nutritional status of children, in addition to enhancing the existing safety net programs.

## Supporting information

**S1 Table. Prevalence of malnutrition among under-five children.**
(PDF)

## Acknowledgments

We would like to express our heartfelt gratitude to Professor Amber Peterman/International Food Policy Research Institution, for helping us in providing "Strengthen PSNP4 Institutions and Resilience (SPIR) Development Food Security Activity (DFSA)" data set. We are thankful to Hawassa University for giving a chance to enroll in a PhD program in Applied Statistics.

## Author contributions

**Conceptualization:** Dafa Duge Wachifo, Zeytu Gashaw Asfaw.

**Data curation:** Dafa Duge Wachifo.

**Formal analysis:** Dafa Duge Wachifo.

**Investigation:** Dafa Duge Wachifo, Zeytu Gashaw Asfaw.

**Methodology:** Dafa Duge Wachifo, Zeytu Gashaw Asfaw.

**Supervision:** Dereje Danbe Debeko, Zeytu Gashaw Asfaw.

**Visualization:** Dafa Duge Wachifo.

**Writing – original draft:** Dafa Duge Wachifo, Zeytu Gashaw Asfaw.

**Writing – review & editing:** Dafa Duge Wachifo, Dereje Danbe Debeko, Zeytu Gashaw Asfaw.

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
