## [Decision Letter · Decision Letter 0]

11 Feb 2025

Dear Dr. Wachifo,

We look forward to receiving your revised manuscript.

Kind regards,

Xiaohong Li

Academic Editor

PLOS ONE

**Journal Requirements:**

**Additional Editor Comments:**

1. The background section of the article needs to be modified. 1) Certain repetitive expressions should be simplified, such as the harm of undernutrition to children's growth and development. 2) Please simplifying and merging the first and second paragraphs into one paragraph in the background. 3) The website is not recommended to be shown in the third paragraph and can be used as a reference. 4) Please explain the statement 'The CIAF of children has a transient nature, which needs to be addressed using a multi-state model.' 5) I don't quite understand the theoretical and practical basis of this study. The author listed many studies in detail in the last section, but it seems that the connection to the theme of this study is not so direct. I personally strongly agree with the author's viewpoint of that more research is required to determine the effects of improved food diversity on children's nutritional status and to examine the intervention's consequences. However, this study did not directly address this issue. Please elaborate on the theoretical and practical basis that is more directly related to the purpose of this research. For example, regarding the topic of changes in nutritional status and influencing factors, what problems have been solved in previous research, what problems have not been solved, and what social significance does solving this problem have.

2. Methods section: 1) Please show some details about the process of sampling. 2) I am very confused about what the authors want to say by citing many references when describing the independent variables. Is the selection of independent variables based on these references, or are the definitions of these independent variables referenced from these literature sources? 3) Please describe the specific version of the statistical software, including R and SPSS. You should even specify which R package program was used.

3. The results section needs to be streamlined and reorganized. 1) There are so many tables and figs shown in the results section, so that the result appears to be without focus. Here are my some suggestions. Table 2 can be presented as an appendix and briefly described in the main text. The results section should present the basic characteristics of the sample at first to the readers. Table 4 and Table 5 can be merged into one table. Figures 3 and 4 provide too little information for readers. In other words, readers cannot interpret useful information from the Figures. Delete these two figs. If the authors think they are very important, they can be attached in the appendix. 2) The description of Figure 5 needs to be modified. What kind of result does Figure 5 want to express. 3) In the table 4, "Spouse or partner is a member of the hh?", what's meaning for 'hh'? 4) I'm not sure what kind of result the author wants to describe in Figure 6? Many of the results statement described are the author's speculations and should not be included in the results. The description of the results should be objective. 5) The results of Table 10 are difficult to be understood. For example, a child at normal only state has probability 0.71 being in under-nutrition in the next , and 0.29 being in over-failure, What is the probability of the state remaining unchanged? The author should provide the total probability of transitioning to the next state (not including unchanged state).

Reviewers' comments:

Reviewer's Responses to Questions

**Comments to the Author**

1. Is the manuscript technically sound, and do the data support the conclusions?

Reviewer #1: Yes

Reviewer #2: Yes

2. Has the statistical analysis been performed appropriately and rigorously?

Reviewer #1: Yes

Reviewer #2: Yes

3. Have the authors made all data underlying the findings in their manuscript fully available?

Reviewer #1: Yes

Reviewer #2: Yes

4. Is the manuscript presented in an intelligible fashion and written in standard English?

Reviewer #1: No

Reviewer #2: Yes

**Reviewer #1:**  Thanks for the opportunity to review this manuscript.

The study analyzes transition rates, durations, and factors influencing the movement between different nutrition states using data from three survey rounds conducted by the International Food Policy Research Institute.

I suggest to put multi state among the keywords

Abstract

In the method section there is a minor grammatical issue in "The institute was performed the three consecutive follow-up surveys rounds." Consider revising to "The institute conducted three consecutive follow-up survey rounds."

The results section is detailed, but some of the findings could be presented more concisely. For example, the phrase "The results indicated that the probability being in the under-failure state in the next time for a healthy infant is higher (0.71)" could be revised to "The probability of a healthy infant transitioning to the under-failure state is high (0.71)."

Introduction

- The introduction is lengthy and could benefit from being more concise. Consider breaking it into shorter paragraphs to enhance readability and separating the background information from the study's objectives.

- There are repeated references to data from UNICEF, WHO, and other organizations. Condense the references to these organizations by summarizing the main points.

- The study's objective is introduced at the end of a very long paragraph. Consider moving it earlier for better context and rephrasing for simplicity.

Research method

- The detailed description of the IFPRI data collection process, including the virtual meeting dates and contacting procedures could be simplified

- The section describing the inclusion criteria and sample selection is clear but could be streamlined for readability.

- The description of the Multi-state Markov Model is technically dense. Consider breaking it into smaller paragraphs with simple explanations before diving into the complex equations.

- Under independent variables, provide a brief rationale for why each variable was selected. This will highlight their relevance to the study.

Equations

- The equations are not consistently formatted and use a mixture of symbols, parentheses, and notation that makes them hard to follow.

- Some symbols are not clearly defined before they are used. For instance, " ℎ ( )" appears in the equations without a prior explanation of what each component represents.

- The text mentions " ( )" as part of the multi-state model but does not define what " " specifically represents (e.g., the state of the child's nutritional status).

- The equation “(3)” is introduced, but there seems to be a mismatch between the equation numbering and the accompanying text, making it confusing to follow the derivation or steps.

- Each equation is presented without sufficient explanation of its purpose or the meaning of the symbols.

- When equations are introduced, provide a sentence that connects the mathematical expression to the research context (e.g., how the transition rates are used in analyzing children’s nutritional states).

Results

- I suggest to Move the interpretation of key findings (e.g., transition probabilities, covariate effects) into the discussion section, where you can explore the implications more thoroughly.

- The statement "Indexes in all rounds were much greater than the current national estimates" is valuable but would benefit from specifying what the national estimates are for comparison.

- For each table, provide a brief, clear interpretation of its data. For example,

- While the results provide detailed statistical outputs, there is a need to better link these findings back to the study's objectives. For example, the study aims to understand the transition rates, durations, and drivers of under-nutrition states. Explicitly stating how the results address these goals would clarify the narrative and emphasize the study's contributions.

Discussion

- Organize the discussion around key themes or findings, such as the prevalence of malnutrition, transition probabilities, and the influence of covariates. This structure will help readers easily grasp the main points

- the finding that female children are less likely to transition to under-nutrition is mentioned but not fully explained. Including possible reasons, such as differences in care or cultural practices, would provide more context and depth.

- The explanation of probabilities and transition rates is often overly technical. While it's important to be precise, translating these findings into real-world implications (e.g., the risk and duration of under-nutrition in children) would make the discussion more accessible and impactful.

- the statement, "This result is consistent with (Oduro et al., 2024) but it is in contrast with (Häkkänen et al., 2020; Hu et al., 2022; Moreira et al., 2019)" does not explain why there might be differences between these studies. Discussing possible reasons for these inconsistencies.

- The discussion section focuses heavily on the Composite Index of Anthropometric Failure (CIAF) rather than exploring the full potential and unique insights provided by the multi-state model. While the CIAF is an essential part of the study, the use of a multi-state Markov model is one of the study's most novel and complex aspects, and it deserves more thorough discussion.

Generalcomment

Fix grammatical error

**Reviewer #2:**  In the introduction section, it is explained that Ethiopia faces high burden of under-nutrition prevalence, ranking among 15 worst affected nations, perhaps short-term or long-term targets in that field in Ethiopia can be explained as a reference.

While the prevalence of malnutrition in Ethiopia is discussed, could comparisons to neighboring countries or regions add further perspective to the problem?

The methodology mentions anthropometric measurements as a key data source. How were these measurements standardized across different survey rounds, and what steps were taken to minimize measurement errors?

The study uses a continuous time Markov multistate model. Could the authors briefly describe why this model was chosen over other statistical models.

The results indicate that children under 5 spend 75% of their time in under-failure states. Could the authors elaborate on why recovery times are so long and suggest factors that may contribute to this? ¬

In the results section, it is explained that Girl had 1.824 times higher likelihood of recovering from an over failure state and less likely to go from a healthy state to an under-failure state compared to boy, perhaps the factors influencing this can be mentioned.

The authors can strengthen the practical recommendation section by elaborating on specific interventions for policymakers. For example, what types of maternal education programs would be most effective?

In the discussion section, it is explained that the child being live in Oromia region is less likely transit from healthy state to malnutrition as compared with child in Amhara region, perhaps the factors influencing this can be mentioned.

Considering the long recovery periods highlighted in the study, could the authors propose emergency interventions to prevent children from remaining in under-failure states for extended periods?

In the discussion section, related to the limitations mentioned, it is possible to explain recommendations for further research so as not to encounter similar limitations that can be in line with the recommendations from the research results in the conclusion section.

**Do you want your identity to be public for this peer review?** For information about this choice, including consent withdrawal, please see our Privacy Policy

Reviewer #1: No

Reviewer #2: No

---

## [Author Response · Author response to Decision Letter 1]

27 Mar 2025

Response to Reviewers Comments

Journal: PLOS ONE

Manuscript: Ref: Submission ID: PONE-D-24-37812

Title: Malnutrition of Under-Five Children in Amhara and Oromia Regions, Ethiopia: Continuous Time Markov Multi-State Modeling

Dafa Duge Wachifo, Dereje Danbe and Zeytu Gashaw Asfaw

Dear Editor,

We are grateful to you and the reviewers for taking the time to read our article and offer insightful criticism. The current version may be improved as a result of your insightful and important feedback. The authors have given the feedback great thought, and we have done our best to respond to each and every one. The entire section of the manuscripts has been read through, and a careful revision has been undertaken. We now anticipate a significant improvement in the overall writing quality and readability. Each reviewer issue has been thoroughly read and corrected, rewritten, or rephrased. Our point by point responses are enclosed below.

The point by point response

Journal Requirements:

1. When submitting your revision, we need you to address these additional requirements. Please ensure that your manuscript meets PLOS ONE's style requirements, including those for file naming. The PLOS ONE style templates can be found at

Author Response: We have revised based on PLOS ONE’S style as indicated in both main body and Title page guidelines.

Author Response: we all agreed to make data available on acceptance of manuscript.

Author Response: Thank you very much for suggestion we did it as commented.

Additional Editor Comments

1. The background section of the article needs to be modified. 1) Certain repetitive expressions should be simplified, such as the harm of under-nutrition to children's growth and development.

Author Response: we tried to simplify them and have revised it by rewriting as presented page 2 in track change file.

2) Please simplifying and merging the first and second paragraphs into one paragraph in the background.

Author Response: very good suggestion, we have revised it by rewriting and merging paragraph 1, 2 and 3 and presented in page 2 & 3 of track change file.

3) The website is not recommended to be shown in the third paragraph and can be used as a reference.

Author Response: Good comment, we have improved it by substituting reference with meeting minute report of five UN agencies (Mitchell & Catherine, 2023).

4) Please explain the statement “The CIAF of children has a transient nature, which needs to be addressed using a multi-state model.”

Author Response: we revised it by elaborating like: The CIAF of children reflects a transient or dynamic nature which needs to be analyzed using a multi-state model. That is, children’s nutritional status can change over time due to a variety of factors such as dietary changes or intervention, health condition, natural and manmade random shocks (drought, locust infestation, pandemic, and war so on). CIAF is composite measure that aggregates the different status of malnutrition (stunting, wasting, underweight and over-weight) and can be used to track the overall malnutrition. A child’s status can move from one CIAF state to another or recovery back healthy state, making it a dynamic. Thus multi-state model allows for understanding of how children might move between different states of CIAF ( under failure, healthy, over failure or remission to these states).

5) I don't quite understand the theoretical and practical basis of this study. The author listed many studies in detail in the last section, but it seems that the connection to the theme of this study is not so direct. I personally strongly agree with the author's viewpoint of that more research is required to determine the effects of improved food diversity on children's nutritional status and to examine the intervention's consequences. However, this study did not directly address this issue. Please elaborate on the theoretical and practical basis that is more directly related to the purpose of this research. For example, regarding the topic of changes in nutritional status and influencing factors, what problems have been solved in previous research, what problems have not been solved, and what social significance does solving this problem have.

• Authors Response: Thank you for comments and question asked for, we tried to revise based on question and presented at page 3 to 5 in track change file as “Previous studies have attempted to identify determinants of under-nutrition using anthropometric measures such as stunting, wasting or underweight separately, but they neither aggregate the indexes nor consider its transient or dynamic nature ( Mohammed & Asfaw, 2018; Amare et al., 2019; Fenta et al., 2020; Belay et al., 2023; Bitew et al., 2021; Raru et al., 2022). Comparing stunting, under-weight, and wasting separately cannot provide a single conclusion on the overall burden of malnutrition among children. Composite index anthropometry failure (CIAF) is preferable to address malnutrition, as WHO criteria and Waterlow's classification which do not discriminate between distinct conditions of indices of malnutrition, (Kuiti & Bose, 2018). The studies conducted by (Fenta et al., 2021a; Kassie & Workie, 2020) in Ethiopia tried to indentify determinants of under-nutrition using composite index; but they did not consider dynamic characteristics of the response nor include overnutrition in their composite index. More recently, Oduro et al., (2024) had studied dynamic characteristics of stunting by using longitudinal data at Nairobi, Kenya and applied multi-state model but they did not include other indicators of malnutrition. As to authors’ knowledge, the studies conducted so far have not fully integrated CIAF in a way that it includes neither under-nutrition and over-nutrition components, nor the dynamic nature of the malnutrition.

2. Methods section: 1) Please show some details about the process of sampling.

Author Response: very good suggestion, were revised as “initially, 196 kebeles from Amhara (115 kebeles) and Oromia (81 kebeles) were selected from 15 productive safety net program (PSNP) implemented districts. However, four kebeles (3 from Amhara and 1 from Omromia) regions were dropped due to two kebeles had no PSNP clients and two kebeles were experienced ongoing civil unrest. Thus, the evaluation sample comprises 192 kebeles. From each kebele 18 households were randomly sampled and final samples those satisfied inclusion criteria were 3,314 households. The inclusion criteria for the sample were that households had to be a PSNP client household, had to have at least one child aged 0–35 months at baseline to have child with under-five age after 2 years follow up, and had to have the mother or primary female caregiver of the 0–35-month-old child as a member of the household.”

2) I am very confused about what the authors want to say by citing many references when describing the independent variables. Is the selection of independent variables based on these references, or are the definitions of these independent variables referenced from these literature sources?

Author Response: NO. Definitely right question and comment; it was technical error of author we revised it by removing all references listed there.

3) “Please describe the specific version of the statistical software, including R and SPSS. You should even specify which R package program was used.”

Author Response: very good suggestions, we revised it like: Data clearance and analysis were performed by SPSS version 27, and R version 4.3.2 with msm 1.7.1. The standard scores of all anthropometric measure indexes were computed by zscorer 0.3.1 packages.

6) The results section needs to be streamlined and reorganized. 1) There are so many tables and figs shown in the results section, so that the result appears to be without focus. Here are my some suggestions. Table 2 can be presented as an appendix and briefly described in the main text. The results section should present the basic characteristics of the sample at first to the readers. Table 4 and Table 5 can be merged into one table. Figures 3 and 4 provide too little information for readers. In other words, readers cannot interpret useful information from the Figures. Delete these two figs. If the authors think they are very important, they can be attached in the appendix.

Author response: We accept all editor suggestions as they are because we are agree to minimize number of Tables and figures from main body of manuscript without sacrificing much information. Since, Tables 4 & 5 are revealed the same output but author had separated it in to two as Table 4 and Table 5 to avoid long Table which is appearing more than one page. So, it can be merged in to one as suggested. Again Figures 3-5 indicate that the movements of children’s nutritional status (CIAF) among 3 specific states. They showed that there were transitions from each state or transitions are bidirectional. That is specific child can transits forward or backward among any of 3 states during study period. Their difference was number of children in process. For instance, in Figure 3, we had plotted transitions of sampled children all together, in Figure 4, we had plotted male and female separately and in Figure 5, we had plotted 4 unique selected children. Therefore, if we delete Figures 3 & 4 and retain Figure 5 as suggested by editor; It is very good insight, here we will benefit with précising the idea. Hence, all suggestions were accepted and improved as shown in track change file from page 9-14.

2) The description of Figure 5 needs to be modified. What kind of result does Figure 5 want to express.

Authors Response: We have modified the interpretation of Figure 5 and presented in track change file page 14. It shows the same things as Figure 3 & 4 but transitions in it were more clear and simple to visualize since sample is small. It indicates that the transition of nutritional status (CIAF) of children or its dynamic characteristics that is it changes as time passes. For example, a child in normal only state at time t can transit to either under failure state or over failure state or stay at normal only state at t+1.

3) In the table 4, "Spouse or partner is a member of the hh?", what's meaning for 'hh'?

Authors Response: In Table 4, “Spouse or partner is a member of the hh?” question is right we accepted comment and modified it as “Spouse or partner is a member of the household (hh)?”

4) I'm not sure what kind of result the author wants to describe in Figure 6? Many of the results statement described are the author's speculations and should not be included in the results. The description of the results should be objective.

Authors Response: We have modified the description of Figure 6 as it, revealed that whether the multi-state model is good fit to the data or not. Based on the result the estimated prevalence rate and observed prevalence approximately equal. Hence, model is good fit to the data.

5) The results of Table 10 are difficult to be understood. For example, a child at normal only state has probability 0.71 being in under-nutrition in the next , and 0.29 being in over-failure, What is the probability of the state remaining unchanged? The author should provide the total probability of transitioning to the next state (not including unchanged state).

Authors Response: Table 10, “….The author should provide the total probability of transitioning to the next state (not including unchanged state).” Really, we appreciate editor’s view and question on probability of remain unchanged in the same state. But the theoretical basis for probability in next state is different from probability of transition. Its aim is to answer the question that, if movement of subject/object is must then where he/she is more probable to move? In our case we know that specific child had transition from health state then we want to estimate how more probable he/she transit to either of states. That is healthy child is more probable (0.71) to transit under failure state than over failure state.

Reviewer #1

Thanks for the opportunity to review this manuscript. The study analyzes transition rates, durations, and factors influencing the movement between different nutrition states using data from three survey rounds conducted by the International Food Policy Research Institute. I suggest to put multi state among the keywords

1. Abstract: In the method section there is a minor grammatical issue in "The institute was performed the three consecutive follow-up surveys rounds." Consider revising to "The institute conducted three consecutive follow-up survey rounds." The results section is detailed, but some of the findings could be presented more concisely. For example, the phrase "The results indicated that the probability being in the under-failure state in the next time for a healthy infant is higher (0.71)" could be revised to "The probability of a healthy infant transitioning to the under-failure state is high (0.71). "

Author Response: Thank you very much for your valuable suggestions, we tried to modify each of them. At the beginning we included “Multi-State Model” among key words but latter author excluded after reading general guideline for key words which suggests that not to include words in the title. The guideline that I have read may be view of scholars so; I am happy to include “Multi-State Model” in key words and also we have revised suggestions to improve the abstract.

2. Introduction: The introduction is lengthy and could benefit from being more concise. Consider breaking it into shorter paragraphs to enhance readability and separating the background information from the study's Objectives: There are repeated references to data from UNICEF, WHO, and other organizations. Condense the references to these organizations by summarizing the main points.

- The study's objective is introduced at the end of a very long paragraph. Consider moving it earlier for better context and rephrasing for simplicity.

• Author Response: very good suggestions; we tried to improve them by separating background information into paragraph1 and objectives into paragraph2 at same time tried to minimize reference repletion by summarizing points.

3. Research method: 1) The detailed description of the IFPRI data collection process, including the virtual meeting dates and contacting procedures could be simplified

Author Response: thank you for comment, we revised by moving statement in to last part of next paragraph at page 6 &7 in track change file and simplified as: “By presenting brief objectives of the current study, the authors contacted and requested IFPRI to get access of the data set. Finally, the data set was accessed on May 29, 2023.”

2) The

---

## [Decision Letter · Decision Letter 1]

26 Jun 2025

Dear Dr. Wachifo,

Thank you for submitting your manuscript to PLOS ONE. After careful consideration, we feel that it has merit but does not fully meet PLOS ONE’s publication criteria as it currently stands. Therefore, we invite you to submit a revised version of the manuscript that addresses the points raised during the review process.

**ACADEMIC EDITOR: Please insert comments here and delete this placeholder text when finished.**

Please address the issues further raised by the reviewer. Moreover, here are some other key areas where the manuscript can be improved to enhance its scientific rigor, clarity, and impact:

**1. Abstract Clarity and Conciseness**

**Issue** : The abstract is information-dense and includes some awkward phrasing (e.g., “under failure,” “over failure”), which may be unclear to non-specialists.**Suggestion** :Avoid jargon like “under failure” and “over failure.” Consider usingUndernutrition or Anthropometric failure, instead of Under failureOvernutrition instead of Over failureAnthropometric failure (for CIAF) instead of Under failureNourished / No anthropometric failure instead of Normal only**Example Rewrite for Clarity:****Original** :“Children in the under failure group had higher transition probabilities.”**Revised** :“Children in the undernutrition group (i.e., those with anthropometric failure) had higher transition probabilities.”**Original** :“Over failure children had a lower likelihood of reverting to a normal state.”**Revised** :“Overnourished children (classified as overweight) had a lower likelihood of reverting to a normal nutritional status.”Streamline sentences. E.g., *“The probability of a healthy infant transitioning to the under-failure state is high (0.71)”* can be simplified to *“Healthy infants had a 71% probability of becoming undernourished.*

** 2. Theoretical and Practical Framing**

**Issue** : The rationale for using the Composite Index of Anthropometric Failure (CIAF) and the Markov model is presented, but the articulation of the *research gap* is buried in repetition and general statements.**Suggestion** : Integrate or refer the following relevant articles regarding CIAF

I.Chowdhury M, Billah B, Rashid M, Almroth M, Kader M. Prevalence and factors associated with severe undernutrition among under-5 children in Bangladesh, Pakistan, and Nepal: A comparative study using multilevel analysis. Scientific Reports 2023 Jun 22;13(1):10183.

II.Anik AI, Chowdhury MR, Khan M, Khan TA, Perera NK, Kader M. Urban-rural differences in the risk factors of severe under-5 child undernutrition based on CISAF in Bangladesh. BMC Public Health. 2021 Nov 23;21(1):2147

Create a sharper distinction between what is already known and what is novel in this study.Explicitly state (if so) : *“To our knowledge, no previous study in Ethiopia has used a continuous-time multistate Markov model to analyze both under- and over-nutrition transitions among under-five children using CIAF.”*

** 3. Methods Section**

**Strength** : Ethical procedures and data source are well described.**Improvement needed** :Clarify the logic behind grouping CIAF categories into three states (UF, NO, OF). If possible consider using a flow diagram for clarity.Equations need better formatting and definition. Symbols such as ℎ, , Q(t), etc., should be defined immediately when first introduced.The transition matrix and equations (3) and (5) lack intuitive explanation. A sentence like *“This represents the instantaneous rate at which a child leaves state h and enters state j”* can aid understanding.

**4. Results Presentation**

The results are numerically rich but difficult to digest due to too many tables and minimal synthesis as suggested by reviewers

** 5. Interpretation and Discussion**

**Issue** : Discussion is technically sound but lacks integration of findings into a **real-world context** .**Suggestions** :Use subsections to structure discussion around key findings: e.g., *“Gender Differences,” “Duration in Malnutrition States,” “Impact of Maternal Education”* .When discussing girls’ lower transition to undernutrition, tie findings to Ethiopian sociocultural contexts more carefully — some statements currently seem speculative (e.g., boys are prioritized but receive riskier food).Add more emphasis on the **novelty of using a Markov model** and its practical implications (e.g., forecasting malnutrition trends, guiding resource allocation).

**6. Language and Grammar**

**Issue** : Recurrent grammar and syntax errors, such as:

*“Girl had 1.824 times higher likelihood…”* → should be “Girls had a 1.824 times higher likelihood…”

*“Child who was older than two years of age had more likelihood…”* → should be “Children older than two years were more likely…”

**Suggestion** : A professional language edit is strongly recommended before final acceptance.

** 7. Conclusion and Policy Implication**

**Issue** : The conclusion is assertive but lacks specificity.**Suggestion** :Offer **specific** recommendations. Instead of “enhance adult and maternal education programs,” state what kind: *“Implement targeted nutrition literacy campaigns for mothers through health extension workers.”*

We look forward to receiving your revised manuscript.

Kind regards,

Manzur Kader

Academic Editor

PLOS ONE

Journal Requirements:

Reviewers' comments:

Reviewer's Responses to Questions

**Comments to the Author**

Reviewer #2: (No Response)

Reviewer #3: All comments have been addressed

Reviewer #4: All comments have been addressed

2. Is the manuscript technically sound, and do the data support the conclusions?

Reviewer #2: Yes

Reviewer #3: Yes

Reviewer #4: Yes

3. Has the statistical analysis been performed appropriately and rigorously?

Reviewer #2: Yes

Reviewer #3: Yes

Reviewer #4: Yes

4. Have the authors made all data underlying the findings in their manuscript fully available?

Reviewer #2: Yes

Reviewer #3: Yes

Reviewer #4: Yes

5. Is the manuscript presented in an intelligible fashion and written in standard English?

Reviewer #2: Yes

Reviewer #3: Yes

Reviewer #4: Yes

Reviewer #2: I think the author did not answer some of the questions i asked, including :

1. In the introduction section, it is explained that Ethiopia faces high burden of under-nutrition prevalence, ranking among 15 worst affected nations, perhaps short-term or long-term targets in that field in Ethiopia can be explained as a reference. In this question the author did not answer what i asked.

While the prevalence of malnutrition in Ethiopia is discussed, could comparisons to neighboring countries or regions add further perspective to the problem?

2. The authors can strengthen the practical recommendation section by elaborating on specific interventions for policymakers. For example, what types of maternal education programs would be most effective? in this question i asked the author to give an example of the concrete program

Reviewer #3: Thank you for the opportunity to review the revision of this manuscript. The revised version focuses on addressing critical public health challenges related to child undernutrition in Ethiopia using a continuous-time Markov multistate model. This methodological choice, combined with longitudinal data, contributes meaningfully to understanding the dynamic nature of child nutritional status transitions and the broader implications for targeted interventions.

The respected reviewer has provided thoughtful and insightful feedback, and the author has responded in a structured and reflective manner. Particularly commendable is the reviewer’s question regarding the rationale behind choosing the multistate model, which allowed the author to clarify its advantage over other statistical approaches in capturing intermediate events. Another notable point is the reviewer’s comment on gender-based differences in recovery, which encouraged the author to delve into cultural and biological factors—this reflects a strong level of critical engagement by the reviewer and a well-appreciated elaboration by the author.

While the author has addressed the reviewer’s comments diligently, a couple of aspects could be further considered. For instance, the response to the suggestion on regional comparisons could be improved with quantitative data comparing Ethiopia’s malnutrition rates to neighboring countries like Kenya or Sudan to offer clearer regional context. Additionally, while the discussion around interventions was improved, specific examples of maternal education program formats (e.g., community-based group sessions vs. mobile health interventions) would make the recommendations more actionable.

Overall, the author’s responsiveness and the reviewer’s depth of analysis are both commendable. The paper has improved significantly in clarity and relevance.

Reviewer #4: The manuscript is clearly written and the study is well conceived and methodologically rigorous, offering an important addition to the area of child nutrition by using continuous-time multi-state Markov model rarely used in this area. The application of CIAF in measuring these dynamic nutritional shifts among children under the age of five in Ethiopia is particularly novel and adds practical value to the study. The authors are very responsive to reviewer comments, and much of the writing has been improved for clarity, justification and organization in key sections. Ethical approval is well recorded and data management seems to be appropriate and transparent. If published, this study will provide reasonable information for policy makers to act for malnutrition in vulnerable groups.

**Do you want your identity to be public for this peer review?** For information about this choice, including consent withdrawal, please see our Privacy Policy

Reviewer #2: No

Reviewer #3: **Yes: ** Bushra Akter

Reviewer #4: **Yes: ** Taiyeba Akter

---

## [Author Response · Author response to Decision Letter 2]

28 Jul 2025

Response to Reviewers Comments

Journal: PLOS ONE

Manuscript: Ref: Submission ID: PONE-D-24-37812

Title: Malnutrition of Under-Five Children in Amhara and Oromia Regions, Ethiopia: Continuous Time Markov Multi-State Modeling

Dafa Duge Wachifo, Dereje Danbe and Zeytu Gashaw Asfaw

Dear Editor,

We are grateful to you and the reviewers for taking the time to read our article and offer insightful criticism. The current version may be improved as a result of your insightful and important feedback. The authors have given the feedback great thought, and we have done our best to respond to each and every one. The entire section of the manuscripts has been read through, and a careful revision has been undertaken. We now anticipate a significant improvement in the overall writing quality and readability. Each reviewer issue has been thoroughly read and corrected, rewritten, or rephrased. Our point by point responses are enclosed below.

ACADEMIC EDITOR:

1. Abstract Clarity and Conciseness

• Issue: The abstract is information-dense and includes some awkward phrasing (e.g., “under failure,” “over failure”), which may be unclear to non-specialists.

• Suggestion:

• Avoid jargon like “under failure” and “over failure.” Consider using

• Undernutrition or Anthropometric failure, instead of Under failure

• Overnutrition instead of Over failure

• Anthropometric failure (for CIAF) instead of Under failure

• Nourished / No anthropometric failure instead of Normal only

Example Rewrite for Clarity:

• Original:

• “Children in the under failure group had higher transition probabilities.”

• Revised:

• “Children in the undernutrition group (i.e., those with anthropometric failure) had higher transition probabilities.”

• Original:

• “Over failure children had a lower likelihood of reverting to a normal state.”

• Revised:

• “Overnourished children (classified as overweight) had a lower likelihood of reverting to a normal nutritional status.”

• Streamline sentences. E.g., “The probability of a healthy infant transitioning to the under-failure state is high (0.71)” can be simplified to “Healthy infants had a 71% probability of becoming undernourished.

Authors Response: Thank you very much for your kind suggestions and insightful comments on the abstract. We have accepted all of them and made the improvements accordingly. Both the track changes and clean versions of the revised file are attached.

2. Theoretical and Practical Framing

• Issue: The rationale for using the Composite Index of Anthropometric Failure (CIAF) and the Markov model is presented, but the articulation of the research gap is buried in repetition and general statements.

• Suggestion: Integrate or refer the following relevant articles regarding CIAF

I.Chowdhury M, Billah B, Rashid M, Almroth M, Kader M. Prevalence and factors associated with severe undernutrition among under-5 children in Bangladesh, Pakistan, and Nepal: A comparative study using multilevel analysis. Scientific Reports 2023 Jun 22;13(1):10183.

II.Anik AI, Chowdhury MR, Khan M, Khan TA, Perera NK, Kader M. Urban-rural differences in the risk factors of severe under-5 child undernutrition based on CISAF in Bangladesh. BMC Public Health. 2021 Nov 23;21(1):2147

• Create a sharper distinction between what is already known and what is novel in this study.

• Explicitly state (if so) : “To our knowledge, no previous study in Ethiopia has used a continuous-time multistate Markov model to analyze both under- and over-nutrition transitions among under-five children using CIAF.”

Authors Response: Thank you very much for your valuable suggestions and comments to improve our manuscript. We have carefully considered and accepted your recommendations. The suggested articles provided us with insightful perspectives, which we used to enhance the quality of our writing and address the identified issues. All revisions have been made and are reflected in both the track-changed (on page 3 at last paragraph) and clean versions of the manuscript.

3. Methods Section

• Strength: Ethical procedures and data source are well described.

• Improvement needed:

• Clarify the logic behind grouping CIAF categories into three states (UF, NO, OF). If possible consider using a flow diagram for clarity.

Authors Response: Thank very much again for comment on the logic behind grouping CIAF in to three. The decision to classify children's nutritional status into three broad categories—under-nourished, over-nourished, and nourished (normal)—was guided by the following considerations:

1) Simplification Based on Nutritional Status:

Children's nutritional status can be broadly categorized as normal, under-nourished, or o ver-nourished. In our classification, a child was considered under-nourished or over-nourished if he/she exhibited at least one anthropometric indicator (stunting, wasting, or underweight). That is the corresponding z-score <-2SD and Z-score > +1 SD for over-nutrition. Children not falling into either extreme were considered to have a normal or nourished status.

2) Avoiding Complex and Ambiguous Subcategories:

While it is theoretically possible to define more granular categories—such as severely under-nourished, moderately under-nourished, overweight, and obese—this approach introduces several practical challenges. Many children present with overlapping anthropometric failures (e.g., stunting and overweight; wasting and underweight; stunting, wasting, and underweight simultaneously), making it difficult to assign them unambiguously to discrete severity-based categories. Therefore, a more aggregated approach was adopted to reduce complexity and ambiguity.

3) Modeling Requirements – Application of Multistate Models:

The use of a continuous-time multistate Markov model to analyze the transitions between nutritional states requires a well-defined and manageable number of states. Reducing the classification into three distinct categories facilitates the modeling of dynamic changes in nutritional status over time, ensuring robustness and interpretability of the model outcomes. Graphically, these models are illustrated using diagrams with boxes representing the CIAF states and with arrows presenting possible transition between the states. Here, it was considered the 3-states for child CIAF transition model depicted in Figure 2, and assumed that subjects would be in any given state at time t equal to zero (t=0). The 6 possible transitions were identified by arrows for child CIAF model: 1) from “under-nourished” to “nourished”; 2) from “nourished” to “under-nourished”; 3) from “nourished” to “over-nourished” 4) from “over-nourished to “nourished”, 5) from “under-nourished” to “over-nourished” and 6) from “over-nourished” to “under-nourished”. The child in normal or healthy state at time t can move either to under-nourished (U) state or move to over-nourished (O) state or stay at the same state after time t+1.

Fig 1: Revised and revisited CIAF with nine sections (Kuiti & Bose, 2018)

A=normal status, B=wasting only, C=wasting and underweight, D=stunting + wasting + underweight, E=Stunting + underweight, F=stunting only, Y=underweight only,

G=Stunting + Over-weight, H = Over-weight only

Nourished (N) = A

Under-nourished (U) = B +C + D + E + F+Y

Over-nourished (O) = G + H

• Equations need better formatting and definition. Symbols such as ℎ, Q(t), etc., should be defined immediately when first introduced.

• The transition matrix and equations (3) and (5) lack intuitive explanation. A sentence like “This represents the instantaneous rate at which a child leaves state h and enters state j” can aid understanding.

Authors Response: Thank you very much for your valuable suggestions and comments to improve our manuscript. We have carefully considered all the feedback, accepted the suggestions, and revised the manuscript accordingly as presented in both tack change and clean file at page 10.

4. Results Presentation

• The results are numerically rich but difficult to digest due to too many tables and minimal synthesis as suggested by reviewers

Authors Response: Thank you again for your insightful review. We acknowledge that the MSM output generates numerous important tables. In response to the reviewers’ suggestions, we carefully reviewed all tables and made a concerted effort to reduce redundancy. We merged several tables and moved others to the supplementary material where appropriate during first review. However, we are concerned that further reductions might compromise the clarity and completeness of the findings. We believe the current balance maintains the integrity of the results while improving readability.

5. Interpretation and Discussion

• Issue: Discussion is technically sound but lacks integration of findings into a real-world context.

• Suggestions:

o Use subsections to structure discussion around key findings: e.g., “Gender Differences,” “Duration in Malnutrition States,” “Impact of Maternal Education”.

o When discussing girls’ lower transition to undernutrition, tie findings to Ethiopian sociocultural contexts more carefully — some statements currently seem speculative (e.g., boys are prioritized but receive riskier food).

o Add more emphasis on the novelty of using a Markov model and its practical implications (e.g., forecasting malnutrition trends, guiding resource allocation).

Authors Response: Thank you once again for your valuable suggestions. Your comments provided us with an opportunity to broaden our perspective and significantly improve the manuscript. We have carefully considered all your suggestions and revised the manuscript accordingly. Both a tracked-changes version and a clean version of the revised manuscript have been submitted. Notably, we have made substantial revisions to clarify and strengthen the novelty of the MSM at last part of discussion, strength and limitation part the study.

6. Language and Grammar

• Issue: Recurrent grammar and syntax errors, such as:

“Girl had 1.824 times higher likelihood…” → should be “Girls had a 1.824 times higher likelihood…”

“Child who was older than two years of age had more likelihood…” → should be “Children older than two years were more likely…”

Suggestion: A professional language edit is strongly recommended before final acceptance.

Authors Response: Thank you very much for your valuable suggestions and comments to improve our manuscript. We have accepted all the suggestions and made revisions accordingly. We also carefully reviewed the manuscript to correct grammatical errors to the best of our ability.

7. Conclusion and Policy Implication

• Issue: The conclusion is assertive but lacks specificity.

• Suggestion:

o Offer specific recommendations. Instead of “enhance adult and maternal education programs,” state what kind: “Implement targeted nutrition literacy campaigns for mothers through health extension workers.”

Authors Response: Thank you very much for valuable suggestions. We carefully include all the suggestions and tried to rewrite recommendation section like: GO/NGO/relevant stakeholders could implement targeted literacy campaigns focus on enhancing maternal knowledge and skills related to improve child nutrition, dietary diversity, childcare, hygiene, and family planning. These campaigns should also aim to address and transform cultural beliefs and practices related to feeding habits, especially during fasting periods and to challenge harmful cultural attitudes surrounding gender roles and their impact on family health and well-being. Utilizing health extension workers or other trained health and nutrition professionals can help effectively deliver these messages at the community level. In the long run, Ministry of Health, Ministry of Agriculture, and other relevant stakeholders should strengthen efforts to reduce poverty and improve the nutritional status of children, in addition to enhancing the existing safety net programs.

We look forward to receiving your revised manuscript.

Kind regards,

Manzur Kader

Academic Editor

PLOS ONE

Journal Requirements:

Reviewers' comments:

Reviewer's Responses to Questions

Comments to the Author

1. If the authors have adequately addressed your comments raised in a previous round of review and you feel that this manuscript is now acceptable for publication, you may indicate that here to bypass the “Comments to the Author” section, enter your conflict of interest statement in the “Confidential to Editor” section, and submit your "Accept" recommendation.

Reviewer #2: (No Response)

Reviewer #3: All comments have been addressed

Reviewer #4: All comments have been addressed

2. Is the manuscript technically sound, and do the data support the conclusions?

Reviewer #2: Yes

Reviewer #3: Yes

Reviewer #4: Yes

3. Has the statistical analysis been performed appropriately and rigorously?

Reviewer #2: Yes

Reviewer #3: Yes

Reviewer #4: Yes

4. Have the authors made all data underlying the findings in their manuscript fully available?

Reviewer #2: Yes

Reviewer #3: Yes

Reviewer #4: Yes

5. Is the manuscript presente

---

## [Editor Report · Decision Letter 2]

4 Aug 2025

Malnutrition among Under-five Children in Amhara and Oromia Regions, Ethiopia: Continuous Time Markov Multi-State Modeling

PONE-D-24-37812R2

Dear Dafa Duge Wachifo

We’re pleased to inform you that your manuscript has been judged scientifically suitable for publication and will be formally accepted for publication once it meets all outstanding technical requirements.

Kind regards,

Manzur Kader, Ph.D

Academic Editor

PLOS ONE
---

## [Editor Report · Acceptance letter]

PONE-D-24-37812R2

PLOS ONE

Dear Dr. Wachifo,

I'm pleased to inform you that your manuscript has been deemed suitable for publication in PLOS ONE. Congratulations! Your manuscript is now being handed over to our production team.

Kind regards,

on behalf of

Dr. Manzur Kader

Academic Editor

PLOS ONE